# Multi-Object Hallucination in Vision Language Models

**Xuweiyi Chen**[1,2*]    **Ziqiao Ma**[1*]    **Xuejun Zhang**[1*]
**Sihan Xu**[1]    **Shengyi Qian**[1]    **Jianing Yang**[1]    **David F. Fouhey**[3]    **Joyce Chai**[1]
[1]University of Michigan    [2]University of Virginia    [3]New York University
https://multi-object-hallucination.github.io/

## Abstract

Large vision language models (LVLMs) often suffer from object hallucination, producing objects not present in the given images. While current benchmarks for object hallucination primarily concentrate on the presence of a single object class rather than individual entities, this work systematically investigates multi-object hallucination, examining how models misperceive (e.g., invent nonexistent objects or become distracted) when tasked with focusing on multiple objects simultaneously. We introduce Recognition-based Object Probing Evaluation (ROPE), an automated evaluation protocol that considers the distribution of object classes within a single image during testing and uses visual referring prompts to eliminate ambiguity. With comprehensive empirical studies and analysis of potential factors leading to multi-object hallucination, we found that (1) LVLMs suffer more hallucinations when focusing on multiple objects compared to a single object. (2) The tested object class distribution affects hallucination behaviors, indicating that LVLMs may follow shortcuts and spurious correlations. (3) Hallucinatory behaviors are influenced by data-specific factors, salience and frequency, and model intrinsic behaviors. We hope to enable LVLMs to recognize and reason about multiple objects that often occur in realistic visual scenes, provide insights, and quantify our progress towards mitigating the issues.

## 1 Introduction

Recent advances in large language models (LLMs) have motivated increasing efforts in adapting them for understanding visual semantics, giving rise to a surge of large vision language models (LVLMs) [1, 37, 43]. These models, whether explicitly trained with grounding data [71] or without [31], demonstrate an impressive grounded understanding of visual entities. This motivates a new prompting paradigm based on user-provided visual cues, referred to as *visual prompting* [45, 60, 27, 58, 58]. Despite their promising performances on various downstream applications [36], LVLMs often suffer from *object hallucination* [44, 10, 26], where they produce objects not present in a given image.

Although object hallucination was initially observed in image captioning describing multiple objects [44], current benchmarks for object hallucination primarily concentrate on the presence of a single object class rather than individual entities. These benchmarks either verify if an object class mentioned in the caption can ground to an object in the image [44, 19], or probe the model about the existence of an object class, sometimes with additional attributes or relations to other objects [26, 32]. There are, however, two key limitations with these setups as shown by a case study in Figure 1. First, grounding is not simply one-to-one between objects and classes, but a many-to-many mapping between objects and phrases [20, 34]. For instance, "apples" could potentially correspond to multiple referents in Figure 1, and the model doesn't necessarily need to recognize all of them to provide such a response. Therefore, being able to produce an object that exists in an image does not necessarily

---

*Authors contributed equally to this work, alphabetized by last names.

38th Conference on Neural Information Processing Systems (NeurIPS 2024).

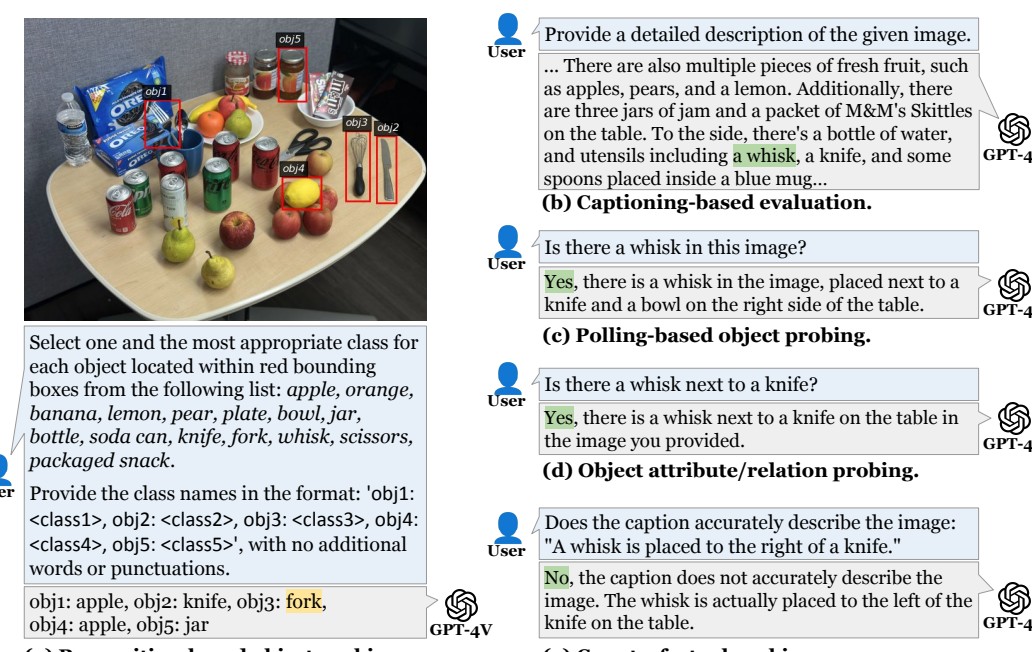

**Figure 1:** A case study that compares our Recognition-based Object Probing Evaluation (ROPE) benchmark with existing benchmarks for object hallucination in GPT-4V. ROPE offers an automated evaluation protocol with controlled output formatting and uses visual prompts to distinctly ground to objects, thus mitigating referential ambiguity. Unlike binary inquiries relying solely on textual descriptions, ROPE challenges the model to identify multiple objects concurrently. We observe that, while GPT-4V can identify the whisk to the left of a knife when prompted about it, the model hallucinates a "fork" when directly tasked to recognize multiple objects.

indicate that the model is free of hallucinations. Second, explicitly instructing the model to recognize multiple objects poses greater challenges compared to simple yes/no inquiries that contain explicit text descriptions for individual objects. For instance, while the model can correctly identify that a whisk is positioned to the left of a knife when "a whisk" is deliberately prompted, as shown in Figure 1(b-d), it may hallucinate a "fork" when directly prompted to recognize both the whisk and the knife (i.e., Figure 1a). This could be due to the common association between knives and forks, which leads to potential hallucinations when models are tasked to focus on multiple objects at the same time. In real-world applications, multi-object querying is crucial for embodied AI tasks. For example, in a cooking scenario, an agent must recognize multiple ingredients and tools simultaneously to be effective. We also present a case study on autonomous driving (see Figure 8 in the appendix), demonstrating how common associations between cars, pedestrians and traffic lights could lead to potential hallucinations. In addition, evaluating multiple objects simultaneously, rather than querying each object individually, can significantly save both time and resources. To enable LVLMs to recognize and reason about multiple objects that often occur in realistic visual scenes and to better quantify the complex phenomena we observed, this paper investigates *multi-object hallucination*, examining how models may misperceive (e.g., by inventing nonexistent objects or becoming distracted) when tasked to focus on multiple objects concurrently, and which factors cause the hallucinations.

We start by introducing Recognition-based Object Probing Evaluation (ROPE) for assessing multi-object hallucination with formatted output control. ROPE features an automated evaluation protocol without black-box neural models or humans as evaluators, and leverages visual prompts to uniquely refer to objects to avoid ambiguity and multiple referents caused by object class names. ROPE considers the distribution of object classes within each image at test time, dividing ROPE into 4 subsets: *In-the-Wild*, *Homogeneous*, *Heterogeneous*, and *Adversarial*. For instance, we investigate scenarios where all tested objects belong to the same class or where each tested object represents a different class. We conduct an in-depth analysis of the hallucination behaviors of LVLMs of different scales and training data (e.g., whether grounding data and conversational data are used), and provide a comprehensive analysis of potential factors that lead to multi-object hallucination. Our main findings are: (1) LVLMs suffer from more hallucinations when tasked to focus on multiple objects, compared

to focusing on a single object; (2) The tested object class distribution affects the hallucination behaviors, revealing that LVLMs may be following shortcuts and spurious correlations; (3) The hallucinatory behaviors of LVLMs are affected by data-specific factors, salience and frequency, and model intrinsic behaviors. These findings provide key insights for the development and application of LVLMs, suggesting for more balanced object distributions, diverse annotations, and enhanced multi-object instructions in grounded LVLMs. We hope this work takes a step towards LVLMs that recognize and reason about multiple objects that often occur in realistic visual scenes.

## 2 Related Work

**Large Vision-Language Models.** There is a growing trend to harness and adapt the powerful large language models (LLMs) for multimodal understanding beyond text [51, 1, 65, 41]. Especially, visual instruction tuning has gained prominence for its competitive performance with a comparatively moderate amount of data and computational resources, leading to a variety of Large Vision-Language Models (LVLMs) [31, 30, 9, 76, 12, 56, 62, 22]. Grounding datasets have been shown to benefit vision-language pre-training [33, 25, 34]. Researchers have developed a family of grounded LVLMs focusing on object grounding to bounding box [40, 8, 72, 3, 63, 70, 39] and segmentation masks [21, 69, 57, 42, 71]. Of the large space of LVLMs, our work is most related to *visual prompting* [58, 60] and *object hallucination* [44, 10]. The paragraphs below describe the two lines of work in detail.

**Visual Prompting.** LVLMs demonstrate their grounded understanding of user-provided visual cues, giving rise to a practical and user-friendly prompting paradigm known as *visual prompting* [58, 60]. Early work on visual prompting in vision-language models can date back to tuning-based methods [2, 61]. Recent studies show that LVLMs demonstrate zero-shot understanding of user-provided visual cues (e.g., a red circle) [45, 60]. This observation allows prompting LVLMs by editing images directly in the pixel space, e.g., by adding visual marks or visual text [27]. Starting from Set-of-Marks (SoM) prompting [58], several training-free methods have been introduced [24, 59, 52]. Recent work further enhances visual prompt understanding by additional visual instruction tuning with diverse visual prompts overlaid on the images [6], or explicitly represent visual pointer tokens in the models [21, 63, 71]. We leverage visual prompting to avoid potential ambiguity in textual descriptions, especially when evaluating multiple object hallucinations for objects of the same class.

**Object Hallucination.** Despite their promising performance on benchmarks, these models frequently generate objects that do not exist in the provided images, a problem known as *object hallucination* [44, 10]. Several methods have been suggested to mitigate the object hallucination issue, such as integrating an external object detector [68], applying visually grounded visual instruction tuning [63, 71] or reinforcement learning [46, 13], performing iterative refinement [75], and adapting the decoding strategies [17]. To quantify progress on mitigating them, various benchmarks have been developed and have revealed the prevalence of object hallucination, even in images that are seen during instruction tuning [68, 29]. We contrast our ROPE benchmark against existing benchmarks and setups in Table 3. ROPE, which is designed for evaluating multi-object hallucination, is distinguished in several ways. First, we deliberately consider the distribution of object classes within a single image at test time. Object hallucination is observed originally in image captioning ap-

| Benchmark | Design Considerations | | | | | #Test |
|---|---|---|---|---|---|---|
| | Multi. | Distr. | Source | Ref. | Eval. | |
| CCEval [68] | ✔ | | Seen | Text | N | 0.1k |
| GAVIE [29] | ✔ | | Mixed | Text | N | 1k |
| FAITHScore [19] | ✔ | | Unseen | Text | N | 2k |
| HaELM [54] | ✔ | | Unseen | Text | N | 5k |
| M-HalDetect [13] | ✔ | | Unseen | Text | H | 0.8k |
| MMHal-Bench [46] | ✔ | | Unseen | Text | N,H | 0.1k |
| CHAIR [44] | ✔ | | Unseen | | A | 46k |
| AMBER [53] | ✔ | | Unseen | Text | A | 1k |
| CIEM [15] | ✔ | | Unseen | Text | A | 5k |
| NOPE [32] | | | Unseen | | A | 3k |
| POPE [26] | | Train | Unseen | | A | 0.5k |
| **ROPE (Ours)** | ✔ | Test | Seen & Unseen | Vis. | A | 5k |

Table 1: An overview of object hallucination benchmarks. For design considerations, we summarize the number of tested images, and if multiple classes and object class distribution (at training and test time) are considered. The image sources include those seen or unseen during instruction tuning. To refer to an object, textual descriptions and visual cues can be adopted. For evaluation, neural models, humans and automatic pipelines are used.

plications, where multiple objects are described [44]. While existing research has demonstrated that the object class distribution in the instruction tuning dataset can influence hallucination patterns [26, 75, 53], the impact of object class distribution within an image at test time remains under-explored. Second, current benchmarks concentrate on the presence of an object class or distinguish instances using textual descriptions like attributes, which can still result in ambiguity and multiple referents. We instead leverage the visual referring prompting setups and use visual cues (i.e.,

marked bounding boxes) to refer to objects. Finally, our evaluation is automated, without black-box neural models or human evaluators.

# 3 Recognition-based Object Probing Evaluation

We introduce the Recognition Object Probing Evaluation (ROPE), an automated protocol for assessing LVLMs in multi-object recognition. ROPE specifically measures object hallucination in VLMs within a multi-object setting, examining how models may misperceive (e.g., by inventing nonexistent objects or becoming distracted) when tasked to focus on multiple objects concurrently, and which factors cause the hallucinations.

## 3.1 Task Setup

**Problem Definition.** To avoid ambiguity from multiple candidate referents when using text prompts, ROPE leverages visual prompts to uniquely refer to objects. ROPE tasks LVLMs with selecting the best matching class for multiple objects, as referred to by the visual prompt, from a predefined set of object classes. Specifically, each sample in the ROPE protocol consists of a quadruple $\{\mathcal{I}, \mathcal{L}, \langle p_1, \cdots, p_n \rangle, \langle o_1, \cdots, o_n \rangle\}$: (1) an image $\mathcal{I}$ consisting of at least $n$ objects; (2) a natural language instruction $\mathcal{L}$ that specifies the recognition task, including $N$ candidate object classes $c_1, \cdots, c_N$; (3) $n$ visual prompts $p_1, \cdots, p_n$, each queries an object in the image; and (4) $n$ object classes $o_1, \cdots, o_n$ as the answers. In this work, we construct a dataset with $N = 50$ and $n = 5$, i.e., models are tasked with recognizing 5 objects out of 50 candidate object classes. Although we use this dataset as an example, ROPE can be applied to any dataset containing multiple objects and their bounding boxes.

**Language Instruction Prompts.** For a fair comparison that accommodates both open-weight and API-based LVLMs, ROPE explicitly instructs models to generate a formatted output of object classes, e.g., `obj1:<class1>, ..., obj5:<class5>` (Figure 2). This format enables automated evaluation through simple parsing. This format enables automated evaluation through simple parsing, avoiding black-box neural models or human evaluators With different analytical purposes, we designed 3 types of task prompts for *Multi-Object* queries, as illustrated in Figure 7 and described as follows.

- *Default*: We probe the model to recognize the 5 objects referred to by the visual prompts concurrently in a single turn of prompting. This setting tasks the model with focusing on and recognizing all 5 objects simultaneously, aiming to capture the complexity involved when the model generates language that includes multiple objects.
- *Student-Forcing*: One potential confounder in the default setting is the model's ability to generate data in the specified format. To separate out errors due to following instructions, we force the model to follow the format template and decode only the object tokens for each of the five objects. Ideally, this setting allows the model to focus solely on object recognition.
- *Teacher-Forcing*: This setting eliminates cumulative error, allowing the model to condition on the correct previous context when generating object classes, leading to upper bound performance in multi-object recognition. We similarly force the model to follow the provided template and decode only the object tokens for each of the five objects, but we replace the previously generated object tokens with the ground truth. This essentially follows the few-shot in-context learning setting. Teacher forcing helps especially when LVLMs take shortcuts by repeating the object class list as ordered in the prompt (e.g., LLaVA-7B [31] and Gemini 1.0 Pro [49] in Figure 2).

For comparison, we also designed task prompts for *Single-Object* query. We probe the model to recognize the object referred to by the visual prompts one at a time, repeating this as 5 independent and individual prompts. Unlike *Default* multi-object query, the model only needs to focus on one object, which can be seen as an extension of the POPE [26] setup from yes/no polling to classification. We refer to Appendix A.1 for the prompt templates for each type of task prompt.

## 3.2 Dataset Construction

**Data Sources and Curation.** Since our goal is to evaluate and analyze multi-object hallucination, the image data must contain multiple objects of diverse classes with instance-level semantic annotations. We build our dataset upon existing panoptic segmentation datasets, including MSCOCO-Panoptic [28, 5] and ADE20K [74], to ensure access to all object instances and their semantic classes. We note that one can build a dataset using the ROPE protocol with any dataset containing multiple objects and their bounding boxes. We describe the data curation pipeline in Appendix A.1.

**Splits by Query Distributions.** As shown in Figure 2and 3, our initial observations indicate that LVLMs are less likely to hallucinate objects when they are tasked with recognizing the same object

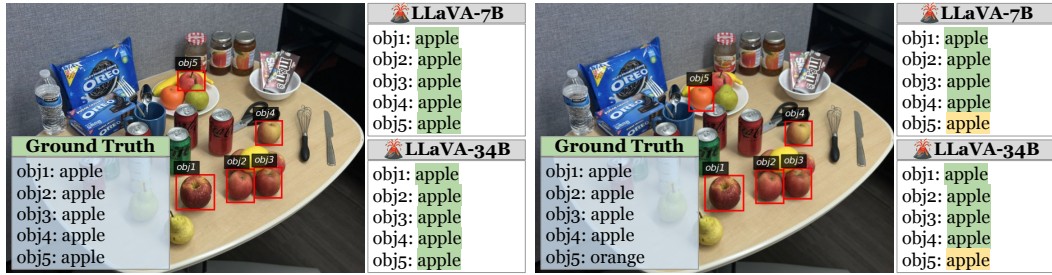

| | **⑤ GPT-4V** | **◆ Gemini 1.0 Pro** | **Qwen-VL-Chat** | **🌋 LLaVA-7B** |
|---|---|---|---|---|
| | obj1: apple | obj1: apple | obj1: apple | obj1: apple |
| | obj2: knife | obj2: orange | obj2: lemon | obj2: orange |
| | obj3: fork | obj3: banana | obj3: bottle | obj3: banana |
| | obj4: apple | obj4: lemon | obj4: packaged snack | obj4: lemon |
| | obj5: jar | obj5: pear | obj5: jar | obj5: pear |

**Ground Truth**
obj1: fork
obj2: knife
obj3: whisk
obj4: lemon
obj5: jar

| | **⑤ GPT-4O** | **◆ Gemini 1.5 Pro** | **Qwen-VL-Max** | **🌋 LLaVA-34B** |
|---|---|---|---|---|
| | obj1: packaged snack | obj1: fork | obj1: packaged snack | obj1: apple |
| | obj2: knife | obj2: knife | obj2: knife | obj2: apple |
| | obj3: whisk | obj3: whisk | obj3: soda can | obj3: apple |
| | obj4: lemon | obj4: lemon | obj4: lemon | obj4: apple |
| | obj5: jar | obj5: jar | obj5: jar | obj5: pear |

Figure 2: A heterogeneous ROPE sample tested with *Deafult* multi-object query, where each of the 5 objects belongs to different object classes. We label the output class as either correct or hallucinated.

**Ground Truth**
obj1: apple
obj2: apple
obj3: apple
obj4: apple
obj5: apple

| **🌋 LLaVA-7B** | | **🌋 LLaVA-7B** |
|---|---|---|
| obj1: apple | | obj1: apple |
| obj2: apple | | obj2: apple |
| obj3: apple | | obj3: apple |
| obj4: apple | | obj4: apple |
| obj5: apple | | obj5: apple |

| **🌋 LLaVA-34B** | | **🌋 LLaVA-34B** |
|---|---|---|
| obj1: apple | | obj1: apple |
| obj2: apple | | obj2: apple |
| obj3: apple | | obj3: apple |
| obj4: apple | | obj4: apple |
| obj5: apple | | obj5: apple |

**Ground Truth**
obj1: apple
obj2: apple
obj3: apple
obj4: apple
obj5: orange

Figure 3: A homogeneous ROPE sample, where the 5 objects belong to the same object class, and a corresponding adversarial ROPE sample, where the last object belongs to a different object class.

class multiple times. However, they tend to make more mistakes when all tasked object classes are different or when a new object class is introduced after multiple repeated tasks. We thus consider the distribution of object classes within each image at test time, dividing ROPE into 4 subsets: *Homogeneous*, *Heterogeneous*, and *Adversarial*, *In-the-Wild*.

- *Homogeneous*: All the 5 tested objects are of the same class, e.g., *AAAAA*.
- *Heterogeneous*: All the 5 tested objects are of different classes, e.g., *ABCDE*.
- *Adversarial*: The first 4 tested objects are of the same class while the last is different, e.g., *AAAAB*.
- *In-the-Wild*: A subset with mixed object class distribution, where the 5 tested objects are randomly chosen and ordered given a test image.

**Attending to Data Contamination.** While data contamination has been explicitly handled in most of the existing benchmarks, object hallucination has been observed even in images that appear in the instruction tuning dataset, such as Visual Genome [68, 29]. To evaluate whether multi-object hallucination can be observed in both seen and unseen images, and to critically determine if training on these images helps reduce hallucinations, we explicitly split our dataset into *Seen* and *Unseen* based on the original split of the datasets.[2] Depending on the object query distributions (4 splits) and whether the image appears in the training split (2 splits), we divide the test into 8 folders.

## 4 Experiments and Results

### 4.1 LVLM Baselines

The proposed ROPE framework, in principle, applies to all LVLMs that can follow format instructions and understand multiple visual prompts. To cover a variety of LVLMs of different scales and training data (e.g., whether grounding data and conversational data are used), we selected the following LVLMs as baselines.

- LVLMs with base LLMs at different scales: LLaVA v1.6 (7B/13B/34B) [31, 30] and Yi-VL (6B/34B) [64].

---

[2]We believe this approach is the best practice, but we also acknowledge that the distinction between seen and unseen images may not be strict. Uncurated web images often overlap with public test images, and researchers have no transparent access to the datasets used to train some of these LVLMs unfortunately [11].

| Models | Default Multi-Object | | | Student-Forcing | | | Teacher-Forcing | | | Single-Object | | |
|---|---|---|---|---|---|---|---|---|---|---|---|---|
| | Wild | Hom. | Het. | Wild | Hom. | Het. | Wild | Hom. | Het. | Wild | Hom. | Het. |
| *Seen* | | | | | | | | | | | | |
| Yi-VL-6B | 2.95 | 5.65 | 1.99 | 3.44 | 6.80 | 3.78 | 5.45 | 26.25 | 4.36 | 0.19 | 0.30 | 0.13 |
| Yi-VL-34B | 8.50 | 15.35 | 3.33 | 8.97 | 16.30 | 4.23 | 10.09 | 19.75 | 4.94 | 0.22 | 2.60 | 0.13 |
| LLaVA-7B | 31.29 | 67.50 | 8.00 | 31.28 | 67.25 | 11.22 | 31.49 | 92.15 | 12.37 | 35.32 | 62.35 | 17.37 |
| LLaVA-13B | 31.54 | 67.63 | 12.64 | 31.49 | 73.25 | 11.54 | 34.97 | 94.25 | 16.03 | 43.13 | 80.60 | 23.91 |
| LLaVA-34B | 39.95 | 85.75 | 18.85 | 52.75 | 85.20 | 33.91 | 56.41 | 95.81 | 25.31 | 55.05 | 86.50 | 18.97 |
| Qwen VL | 2.73 | 6.60 | 1.03 | 6.25 | 16.00 | 3.65 | 18.74 | 71.50 | 5.45 | 8.73 | 16.05 | 5.58 |
| Qwen VL-C | 8.72 | 16.90 | 6.67 | 5.26 | 8.60 | 4.10 | 12.11 | 47.75 | 8.08 | 25.99 | 43.40 | 13.21 |
| CogVLM | 0.04 | 0.00 | 0.00 | 0.00 | 0.00 | 0.00 | 0.10 | 0.95 | 0.00 | 0.00 | 0.00 | 0.00 |
| CogVLM-G | 0.00 | 0.00 | 0.00 | 9.86 | 13.50 | 6.79 | 22.64 | 75.45 | 0.45 | 11.25 | 22.65 | 7.12 |
| CogVLM-C | 12.89 | 22.75 | 7.18 | 25.37 | 43.63 | 12.03 | 28.25 | 72.80 | 17.50 | 30.16 | 56.00 | 16.35 |
| LLaVA-7B* | | N/A | | 9.16 | 16.40 | 5.51 | | N/A | | 11.68 | 23.55 | 9.36 |
| GLaMM* | | N/A | | 27.11 | 53.35 | 13.01 | | N/A | | 63.81 | 81.75 | 53.40 |
| GroundHOG* | | N/A | | 23.57 | 30.80 | 24.23 | | N/A | | 44.80 | 43.10 | 38.97 |
| IDEFICS | 0.00 | 1.45 | 0.13 | 6.25 | 18.70 | 0.64 | 17.37 | 76.15 | 10.06 | 4.62 | 0.00 | 0.32 |
| CogVLM-2 | 21.51 | 37.55 | 17.31 | 37.02 | 70.85 | 12.69 | 37.10 | 73.50 | 17.44 | 21.16 | 38.75 | 13.65 |
| MiniCPM-V | 34.75 | 59.91 | 17.37 | 31.62 | 62.80 | 13.65 | 32.16 | 68.05 | 16.79 | 27.42 | 54.35 | 16.92 |
| GPT-4V† | 53.80 | 77.55 | 40.83 | | N/A | | | N/A | | 55.89 | 78.25 | 41.03 |
| GPT-4O† | 71.27 | 89.25 | 66.03 | | N/A | | | N/A | | 60.77 | 73.92 | 54.31 |
| LLaVA-7B‡ | 21.26 | 52.40 | 7.69 | | N/A | | | N/A | | 30.59 | 60.85 | 12.69 |
| +OPERA | 24.07 | 58.65 | 7.35 | | N/A | | | N/A | | 30.44 | 60.85 | 13.27 |
| *Unseen* | | | | | | | | | | | | |
| Yi-VL-6B | 2.74 | 3.88 | 1.14 | 3.18 | 4.24 | 5.20 | 4.04 | 10.90 | 10.57 | 0.14 | 0.45 | 0.08 |
| Yi-VL-34B | 7.77 | 15.63 | 4.23 | 10.28 | 18.04 | 7.97 | 11.24 | 22.49 | 12.03 | 0.46 | 2.37 | 0.41 |
| LLaVA-7B | 30.56 | 64.12 | 10.33 | 30.55 | 68.16 | 10.24 | 31.89 | 90.33 | 13.25 | 34.88 | 64.41 | 16.18 |
| LLaVA-13B | 27.56 | 63.10 | 8.37 | 27.41 | 63.10 | 8.37 | 35.65 | 91.09 | 14.80 | 42.66 | 71.92 | 23.41 |
| LLaVA-34B | 29.30 | 79.43 | 17.72 | 29.45 | 91.18 | 14.39 | 37.40 | 95.51 | 17.92 | 51.71 | 77.88 | 30.81 |
| Qwen VL | 2.80 | 1.95 | 7.06 | 7.17 | 16.41 | 4.15 | 10.34 | 58.00 | 4.07 | 17.73 | 31.22 | 9.51 |
| Qwen VL-C | 18.86 | 30.73 | 8.78 | 16.16 | 27.80 | 7.72 | 21.81 | 58.00 | 11.14 | 34.20 | 57.31 | 15.37 |
| CogVLM | 0.03 | 0.00 | 0.00 | 0.00 | 0.00 | 0.00 | 0.00 | 0.15 | 0.00 | 0.00 | 0.00 | 0.00 |
| CogVLM-G | 0.00 | 0.00 | 0.00 | 8.20 | 1.47 | 5.77 | 23.82 | 81.20 | 1.81 | 10.32 | 10.74 | 9.11 |
| CogVLM-C | 15.56 | 26.57 | 5.53 | 17.18 | 41.27 | 6.02 | 22.81 | 56.04 | 6.67 | 30.56 | 52.00 | 13.50 |
| LLaVA-7B* | | N/A | | 7.59 | 12.12 | 4.88 | | N/A | | 12.71 | 22.49 | 8.46 |
| GLaMM* | | N/A | | 29.11 | 54.53 | 14.23 | | N/A | | 68.65 | 77.06 | 52.28 |
| GroundHOG* | | N/A | | 23.11 | 24.69 | 26.26 | | N/A | | 40.73 | 30.37 | 38.13 |
| IDEFICS | 0.39 | 0.37 | 0.33 | 9.03 | 24.45 | 2.68 | 24.80 | 83.02 | 7.64 | 4.62 | 3.67 | 6.50 |
| CogVLM-2 | 20.99 | 35.06 | 15.93 | 24.64 | 38.04 | 23.17 | 26.74 | 46.04 | 26.59 | 11.13 | 30.94 | 5.77 |
| MiniCPM-V | 32.96 | 59.92 | 16.60 | 31.77 | 58.98 | 14.15 | 31.87 | 60.98 | 16.34 | 25.56 | 47.76 | 14.39 |
| GPT-4V† | 45.46 | 63.12 | 34.17 | | N/A | | | N/A | | 47.34 | 64.94 | 35.45 |
| GPT-4O† | 63.27 | 80.29 | 54.47 | | N/A | | | N/A | | 63.45 | 79.84 | 53.74 |
| LLaVA-7B‡ | 13.96 | 31.88 | 3.98 | | N/A | | | N/A | | 26.95 | 54.41 | 11.06 |
| +OPERA | 13.20 | 37.14 | 3.82 | | N/A | | | N/A | | 27.90 | 56.69 | 11.22 |

* Mechanistically grounded LVLMs take visual prompts by dedicated pointer tokens. We slightly adapt the text prompt and probe the object classes with the highest probabilities. We also apply such probabilistic probing to LLaVA-7B for comparison, as all of three models adopt Vicuna-7B v1.5 [73] as the base LLM. See Appendix A.1 for details.

† For GPT models, student/teacher forcing doesn't apply as they are API-only.

‡ OPERA is implemented based on LLaVA-7B v1.5.

Table 2: Averaged accuracy of baselines on the *In-the-Wild*, *Homogeneous*, and *Heterogeneous* splits. The **bold** marker denotes the best-performing baseline and the underlined marker denotes the second-best-performing baseline.

- LVLMs with conversational/grounded instruction tuning: QwenVL-Base/Chat (7B) [3] and CogVLM-Base/Chat/Grounding v1.1 (19B) [55].
- Mechanistically grounded LVLMs: GLaMM (7B) [42] and GroundHOG (7B) [71].
- LVLMs with RL-based finetuning: MiniCPM-V [66]
- Other LVLMs: IDEFICS-instruct (9B) [23], MiniCPM-V v2.5 (8B) [16, 67], GPT-4V [37], and GPT-4O [38].

For mechanistically grounded LVLMs that take visual prompts through specially designed mechanisms, such as pointer tokens in GroundHOG [71], we additionally experiment with their default format and report whichever yields higher performance. For other LVLMs, we overlay the visual prompts on the images using a red bounding box with a width of 2 and visual text specifying the object index, presented with a white italic font on a black background with an alpha value of 0.75 for contrast and visibility.

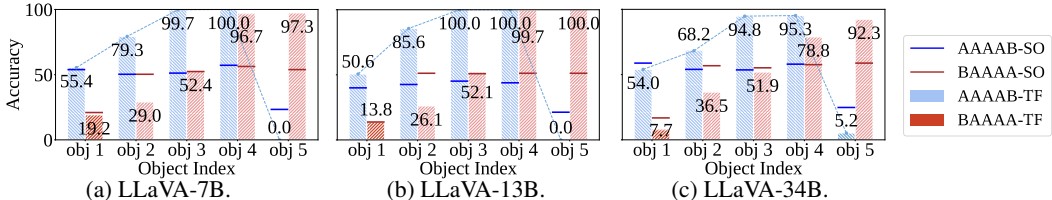

(a) LLaVA-7B.      (b) LLaVA-13B.      (c) LLaVA-34B.

Figure 4: The performance of the LLaVA on the adversarial split, organized by the query sequence of *AAAAB* and *BAAAA*, reveals significant vulnerabilities as the model's accuracy dramatically declines for object 5 in *AAAAB*. **SO** stands for single-object probing and **TF** stands for teacher-forcing probing.

## 4.2 Main Results and Findings

We summarize the average results across the splits in Table 2 and present the most important findings below. The full tables appear in Appendix A.2.

**Multi-object tasks introduce more hallucinations.** Our immediate observation is that LVLMs suffer from more hallucinations when tasked with focusing on and recognizing multiple objects compared to a single object. Across most of the models and test splits, we find that the average accuracy of single-object queries (i.e., probing object classes one at a time) significantly outperforms that of all three types of multi-object queries. The first exceptions are GPT-4O, MiniCPM-V, and CogVLM-2, the latter two leverage LLaMA-3 [35]. Another exception to this is when teacher-forcing is applied to homogeneous test splits, which demonstrates an unreasonably high accuracy. We discuss them later in this section.

**Heterogeneous queries introduce more hallucinations.** We find that for all models and query methods, more heterogeneous queries lead to substantially more hallucinations, with performance decreasing from homogeneous to in-the-wild to heterogeneous test sets. The impact of heterogeneity applies to even start-of-the-art LVLMs like GPT-4O (Figure 2), although this performance gap is more significant in open-weight models.

**Language bias and shortcuts can lead to multi-object hallucinations.** In the teacher-forcing setting, where there are no cumulative errors, LLaVA models score over 90% accuracy. There are three possible hypotheses for this abnormal observation: (1) LVLMs are smart enough to learn object recognition in general through few-shot in-context learning in the teacher-forcing setting; (2) LVLMs learn to recognize one specific object through few-shot in-context learning in the teacher-forcing setting; or (3) LVLMs simply exploit language biases and rule-based shortcuts (e.g., repeating previous answers). To reach a conclusion on this, we examine an *Adversarial* split, in which the first four tested objects are of the same class and we probe an object of a different class for the last one (e.g., *AAAAB*). We compare the single-object query performance with the teacher-forcing performance on the fifth object (object B). We anticipate the following outcomes: If hypothesis (1) is correct, the teacher-forcing performance should outperform the single-object query. If hypothesis (2) is correct, the teacher-forcing performance should perform on par with the single-object query. If hypothesis (3) is correct, the teacher-forcing performance should underperform compared to the single-object query. For a controlled comparison in the multi-object setting, we also reverse the order of queries (i.e., *BAAAA*) and repeat the experiments.

We present the results of LLaVA models on the unseen split in Figure 4, with the full results available in Appendix A.2. We find that the model's predictions on class A progressively improve, scoring nearly perfectly starting from the third repetition. However, the model's performance on the last object (with the different class label B) drops to nearly zero, with almost all hallucinations labeling it as A. This is in stark contrast to 23.35% if these objects are probed individually or 19.16% when these objects are placed as the first to query in multi-object settings. Our findings suggest that hypothesis (3) is true, indicating that the LVLMs' high performance on homogeneous queries could be an illusion resulting from textual shortcuts. We observe that models show lower performance in identifying object class B in single-object analysis, potentially due to the higher salience of object class A in these images.

**Multi-object hallucinations occur in both seen and unseen images.** We finally investigate whether our observations and findings hold uniformly in both seen and unseen splits. We observe that the gap between multi-object hallucination and single-object hallucination, as well as the reliance on shortcuts, persists. Although most of the models perform slightly better on seen images, the trends remain consistent across both splits. While large-scale training is involved in developing

these LVLMs, it appears they might not have fully exploited the fine-grained information in the data. Training on these images does not significantly reduce object hallucinations.

## 4.3 What May Help and What May Not?

Comparing the tested LVLMs, we discuss our observations regarding design considerations that may or may not help reduce multi-object hallucinations.

**Scaling the base LLM: data and parameters.** We find that using base LLMs with more parameters reduces single-object hallucinations, but may not have the same effect on multi-object hallucinations. We observe a consistent increase in performance with larger LLaVA models in the seen set and in single-object queries, but not in the unseen set with multi-object queries. One possible explanation for this finding is that LLMs with more parameters are better at memorizing seen images, as the performance gap between seen and unseen images is also more significant in larger models. We also notice that the performance gap between single-object probing and multi-object probing does not apply to MiniCPM-V and CogVLM-2, which adopt a LLaMA-3 (8B) [35] base LLM pre-trained with 15T tokens, as they fail to follow the instruction sometimes. Compared to LLaVA models developed upon LLaMA-2 (7/13B) [50] and Yi (34B) [64] with 2T and 3T pre-training tokens, these models underperform in quantitative measures due to instruction following error but exhibit greater robustness when multiple visual prompts are presented.

**Visual instruction fine-tuning: chat and grounding.** While it's surprising that conversational tuning reduces multi-object hallucinations, we observe that models without conversational tuning struggle to follow instructions and are prone to shortcuts, such as repeating the list of all object class candidates in order or consistently repeating the first candidate. This might also explain why grounded tuning in CogVLM-G is of little help in reducing multi-object hallucinations thus far. These models typically lack conversational fine-tuning, and there is currently no available grounded dialogue data at scale. While mechanistically grounded LVLMs show strong performances in single-object probing, there remain a gap in multi-object probing with student forcing. This could be attributed to a significant portion of the grounded instruction tuning dataset consisting mainly of short captions or questions featuring one single or few objects.RL-based finetuning approaches, such as MiniCPM-V, demonstrate promising results across diverse settings, surpassing single-object results in both the Wild and Homogenous settings. Upon inspection, we found that this model demonstrates strong visual in-context learning capability and improves correct recognition when objects of the same classes are probed together.

**Decoding and inference time strategy** Decoding algorithms like OPERA introduce nuanced improvements in specific multi-object settings[18]. In default multi-object tasks, OPERA shows marginal performance enhancements for LLaVA-1.5, but its effectiveness declines in tests with greater heterogeneity, to the point it can even lower performance. This suggests OPERA is beneficial in homogeneous contexts but requires further refinement in handling mixed object scenarios effectively.

# 5 Analysis of Hallucinatory Behaviors

## 5.1 Potential Hallucinatory Factors

The task setup described above allows us to evaluate LVLMs in multi-object hallucinations and identify hallucinatory behaviors. Based on existing literature and our case studies, we further identify potential factors that correlate to and potentially explain these hallucinations.

**Data-specific Factors.** We consider the following factors that are specific to the tested sample (e.g., object and token positions), and are not relevant to the frequency distribution.

- *Input Order*: we consider the order in which the object classes are presented in the input prompt containing all candidates.
- *Query Homogeneity*: We define query homogeneity as the total number of task objects of the same class, normalized by the total number of queried objects (five in this work).
- *Object Token Position*: Zhou et al. [75] has shown that more hallucinations occur in the latter part of captions. In this work, the object indices directly correspond to the object token positions.
- *Object Homogeneity*: We define object homogeneity as the number of object types in the image, calculated upon panoptic annotations.
- *Object Centrality*: Previous research has identified a center bias in datasets and models, indicating that objects are disproportionately located at the center of images in detection models [47, 48]. We define object centrality as one minus the distance $d$ between the object's bounding box center and the image center, normalized by the diagonal distance $D$ from the center to the corner.

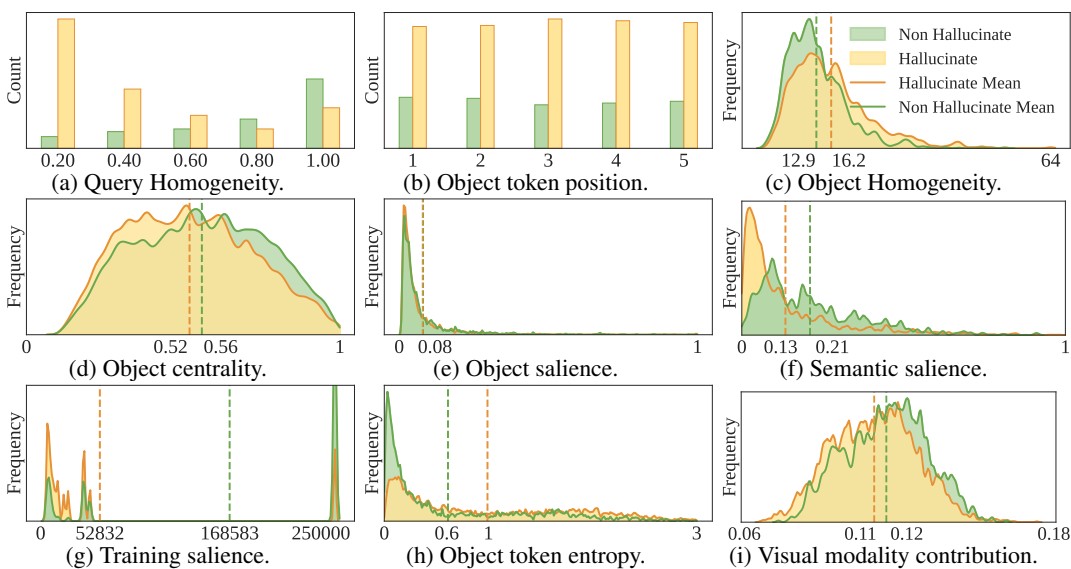

Figure 5: A comparison of the distribution of hallucinatory versus non-hallucinatory object classes in LLaVA-13B, across the unseen split under student forcing.

**Salience and Frequency.** We consider the following factors that are related to the saliency or frequency of the visual object or the object class.

- *Object Salience*: Previous research has shown that smaller objects are harder to detect and ground to [14, 34]. We define object salience as the ratio of the number of pixels occupied by the object's instance segmentation mask to the total number of pixels in the image.
- *Semantic Salience*: We observe and hypothesize that LVLMs are less likely to hallucinate objects when they co-occur with multiple copies of the same class ("jar" in Figure 2). We define semantic salience as the ratio of the total number of pixels in all instances of the same class, to the total number of pixels in the image.
- *Training Salience*: Previous research has shown that spurious co-occurring patterns in the training data can lead to object hallucinations [26, 75]. We use the log frequency of classes in MSCOCO as a proxy for training salience following previous work, and hypothesize that LVLMs tend to hallucinate more on less frequent objects in the training set.

**Model Behaviors.** We consider the following factors relevant to the mechanistic behaviors.

- *Object Token Entropy*: Zhou et al. [75] have shown that object hallucinations are more likely when the decoded object tokens have a higher log perplexity. In our work, we define object token entropy as the entropy of the logits of the first token in the generated word. Given $\mathbf{s}$ as the softmax logits of the generated word's first token, we calculate the entropy using the following formula: $H(\mathbf{s}) = -\sum_i \mathbf{s}_i \log(\mathbf{s}_i)$. Simply put, higher entropy indicates greater uncertainty in the model's prediction for the first token, which can lead to more frequent object hallucinations.
- *Visual Modality Contribution*: We hypothesize that LVLMs pay less attention to the visual modality during object hallucinations. Motivated by the modality importance score [7], we define Visual Modality Contribution (VMC) as the proportion of attention allocated to visual tokens compared to textual tokens. To quantify this, we analyze the attention weights of the last generated token across all heads and layers. The VMC is computed as follows: $\text{VMC} = \sum_{i \in V} \alpha_{ij} / \left( \sum_{i \in V} \alpha_{ij} + \sum_{k \in T} \alpha_{kj} \right)$, where $\alpha_{ij}$ represents the attention weight assigned to visual token $i$ at head $j$, and $\alpha_{kj}$ represents the attention weight assigned to textual token $k$ at head $j$. The sets $V$ and $T$ denote the visual and textual tokens, respectively. By examining the VMC, we can determine how much attention is given to visual inputs in comparison to textual inputs. A lower VMC may indicate a higher likelihood of object hallucinations due to insufficient attention to visual cues.

## 5.2 When Do LVLMs Experience Multi-Object Hallucinations?

In Figure 5, we compare the distribution of these factors between *hallucinatory* and *non-hallucinatory* objects in the student forcing setting on the unseen split using LLaVA-13B. For continuous values, we use ridgeline plots, and for discrete values with fewer bins, we use bar charts.

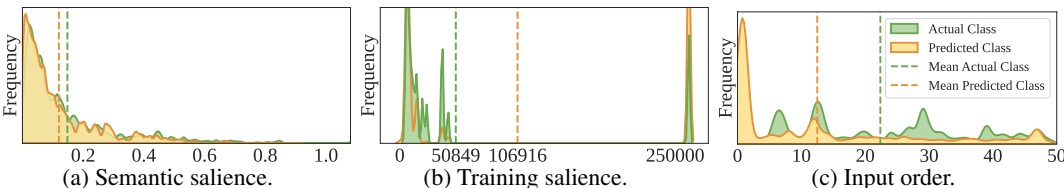

Figure 6: A comparison of the distribution of actual versus predicted object classes for all hallucinatory objects in the student forcing setting on the unseen split using LLaVA-13B.

**Data/Task-specific Factors.** We observed that specific data factors, such as query and object homogeneity, significantly influence model performance, with increased hallucination occurring when models process images featuring multiple object classes or a variety of objects. For positional factors, the position of object tokens seems to have minimal impact and the object centrality has only a slight influence as LVLMs tend to hallucinate objects more frequently when they are positioned away from the center. This tendency may stem from a reporting bias, as objects mentioned in captions are typically foreground objects that distribute toward the centers of images.

**Salience and Frequency.** We note that semantic salience significantly affects the model's performance, as it is more prone to hallucinate an object class that is less salient within the image. Conversely, the salience of individual objects does not statistically correlate with hallucination incidents. This implies that LVLMs may rely more on the presence of co-occurring objects of the same class to predict the labels of queried objects, rather than solely on the presence or salience of the objects themselves. Additionally, training salience plays a crucial role as models are less likely to hallucinate object classes that frequently appear in training.

**Intrinsic Behaviors.** The intrinsic behaviors of the model provide significant insights into its tendencies to hallucinate. Similar to Zhou et al. [75], we find that models are more prone to hallucination when they experience uncertainty or confusion, especially in scenarios involving multiple objects, as evidenced by higher token entropy. Furthermore, the contribution from the visual modality consistently registers below 20%, suggesting that current LVLMs may rely more heavily on linguistic contexts. There is a marginal increase in the likelihood of hallucination when models pay less attention to the visual context.

### 5.3 How Do LVLMs Experience Multi-Object Hallucinations?

In Figure 6, we conducted a detailed comparison of the distribution of actual versus predicted object classes within the context of hallucinatory objects, examining factors such as semantic salience, training salience, and input order. Although semantic salience is a key factor in determining whether a model hallucinates, it appears to have minimal impact on the prediction of hallucinated objects. Our analysis also shows that models are more likely to hallucinate object classes that are prevalent in the training data, but the reverse is not necessarily true. Additionally, there is a notable preference for models to hallucinate objects that are listed early in the input prompt as candidate classes. Overall, our findings indicate that spurious correlations may lead to hallucinations involving multiple objects.

## 6 Discussions and Conclusion

Hallucinations in large vision-language models (LVLMs) can occur at different scales and granularities. In this study, we study the problem of *multi-object hallucination*, examining how LVLMs may misperceive when tasked to focus on multiple objects concurrently, and which factors cause the hallucinations. we introduce Recognition-based Object Probing Evaluation (ROPE), an automated evaluation protocol designed to account for the distribution of object classes within a single image during testing and to use visual referring prompts to reduce ambiguity. Our research provides key insights for the development and application of LVLMs. Since models tend to experience more hallucinations with multiple objects than with single ones, it may be advantageous to probe objects individually in visual prompts to enhance performance. The likelihood of a model's hallucinatory output is linked to various data factors and model behaviors. Particularly in situations involving heterogeneous data and low certainty from the model, there is an increased risk of hallucinations, and users should be vigilant. Moreover, our analysis indicates that merely adopting (grounded) instruction tuning and scaling the base language model may not be enough to fully address the issue of object hallucination. There is a need for more balanced object distributions, annotations of objects away from image centers, and an increase in diversity. Introducing instructions that require multiple visual pointers and complex multi-object reasoning is also crucial.

**Acknowledgement** This work was supported in part by NSF IIS-1949634, NSF SES-2128623, the DARPA Perceptual Task Guidance (PTG) Program, and the DARPA Machine Common Sense Program. Our experiments have also benefited from the Microsoft Accelerate Foundation Models Research (AFMR) program. We thank the Amazon AGI team for GroundHOG model access. The authors would like to thank Yichi Zhang and anonymous reviewers for their valuable feedback.

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

# A Additional Experiments, Results, and Discussions

## A.1 Reproducibility

**Data Curation Pipeline.** Our data curation pipeline involves several essential steps designed to prepare and refine our dataset for evaluating multi-object hallucination. The pipeline begins by filtering images and candidate objects to query. We consider valid objects to be those belonging to the top 50 "thing" classes and exclude objects with a bounding box area less than 1% of the total image area. We discard images containing fewer than 5 valid objects, and allow an intersection-over-union between bounding boxes of no more than 0.1, which preserves data integrity while ensuring high image quality. We apply this pipeline to MSCOCO-Panoptic [28, 5] and ADE20K [74].[3]

| Dataset | Total | COCO | ADE |
|---|---|---|---|
| Wild | 1539 / 1172 | 732 / 547 | 807 / 625 |
| Hom. | 312 / 490 | 168 / 289 | 144 / 201 |
| Het. | 400 / 246 | 200 / 76 | 200 / 170 |
| Adv. | 168 / 334 | 54 / 170 | 114 / 164 |

Table 3: An overview of object hallucination benchmarks. For design considerations, we summarize the number of tested images, and if multiple classes and object class distribution (at training and test time) are considered. The image sources include those seen or unseen during instruction tuning. To refer to an object, textual descriptions and visual cues can be adopted. For evaluation, neural models, humans and automatic pipelines are used.

**Language Instruction Prompt Templates.**
We illustrate the 4 types of task prompts for *Single-Object* and *Multi-Object* queries in Figure 7, and document the prompts below.

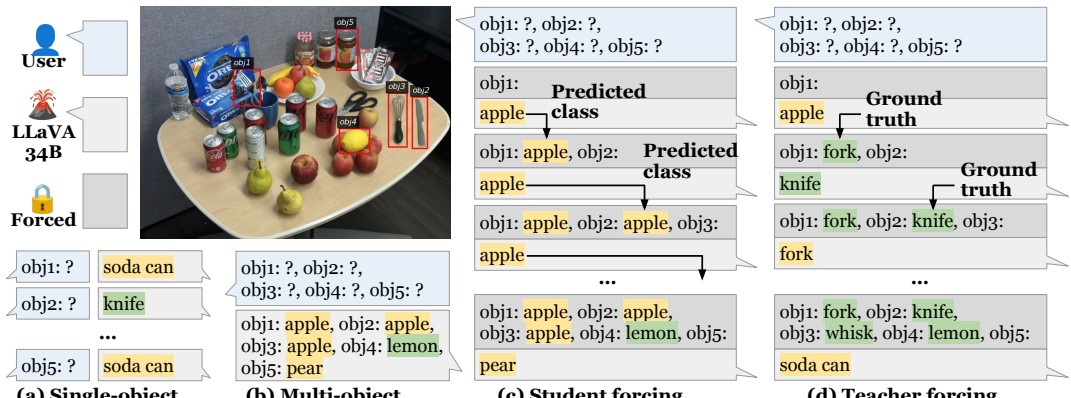

(a) Single-object.    (b) Multi-object.    (c) Student forcing.    (d) Teacher forcing.

Figure 7: Different types of instruction settings of ROPE. In a single turn of prompting without format enforcement, we probe the model to recognize the 5 objects referred to by the visual prompts **(a)** one at a time in the **single-object** setting and **(b)** concurrently in the **multi-object** setting. We further enforce the model to follow the format template and decode only the object tokens for each of the five objects **(c)** without output manipulation in **student forcing** and **(d)** replacing all previously generated object tokens with the ground truth classes in **teacher forcing**.

- Multi-Object Default Probing, Student Forcing, and Teacher Forcing:

  > Select one and the most appropriate class for each object located within red bounding boxes from the following list: [CLASS NAMES]. Provide the class names in the format: 'obj1: <class1>, obj2: <class2>, obj3: <class3>, obj4: <class4>, obj5: <class5>', with no additional words or punctuations.

- Multi-Object Probabilistic Probing:

  > (GroundHOG) Describe object 1 <PTR> and object 2 <PTR> and object 3 <PTR> and object 4 <PTR> and object 5 <PTR>. obj1: <class1>, obj2: <class2>, obj3: <class3>, obj4: <class4>, obj5: <class5>

  > (GLaMM) What are the classes of [<bbox> LIST]? obj1: <class1>, obj2: <class2>, obj3: <class3>, obj4: <class4>, obj5: <class5>

---

[3]Available at `https://huggingface.co/datasets/sled-umich/ROPE`

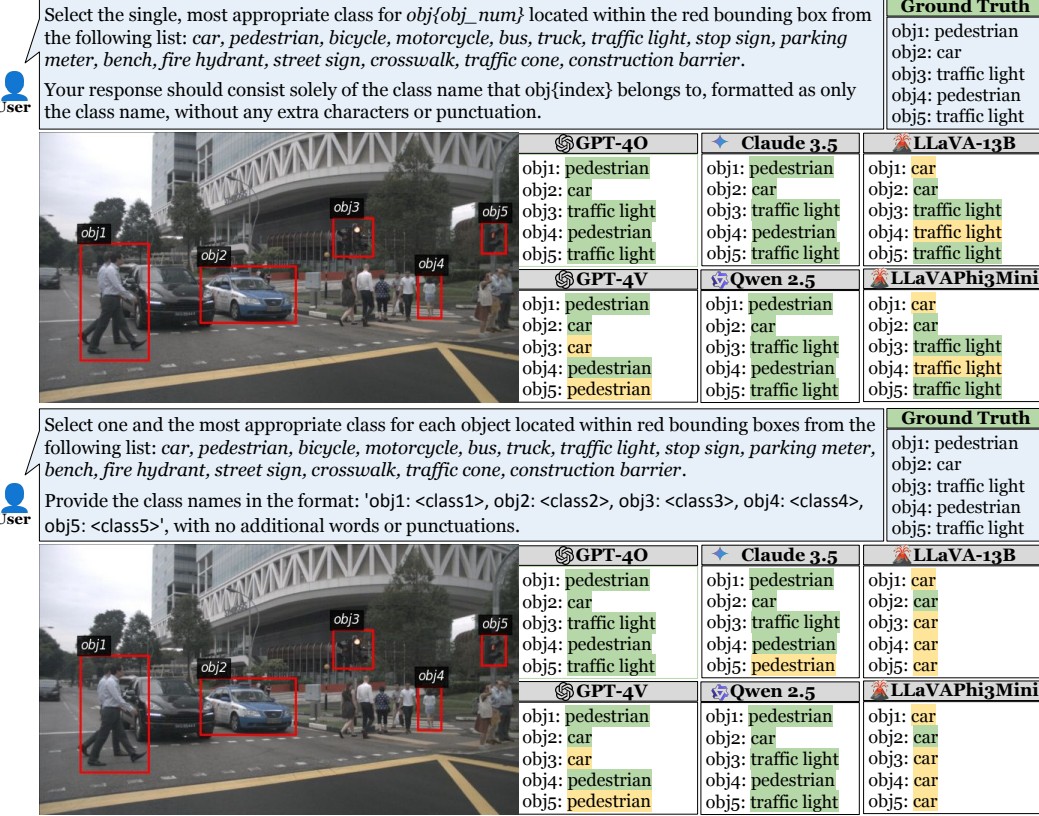

Figure 8: Single and multi-object hallucination under the default setting in nuScenes [4].

> (LLaVA) There are five red bounding boxes in this image. For each object within the red bounding boxes, identify its class. Provide the class names in the format: 'obj1: <class1>, obj2: <class2>, obj3: <class3>, obj4: <class4>, obj5: <class5>', with no additional words or punctuation.

• Single Object Default Probing:

> Select the single, most appropriate class for `obj<obj_num>` located within the red bounding box from the following list: `[CLASS NAMES]`. Your response should consist solely of the class name that `obj<obj_num>` belongs to, formatted as only the class name, without any extra characters or punctuations.

• Single Object Probabilistic Probing:

> (GroundHOG) Describe object `<PTR>` in a word.

> (GLaMM) What is the class of `<bbox>`?

> (LLaVA) Describe the object in the red bounding box labeled `obj<obj_num>` in a word.

**Computational Resources.** Our experiments were conducted on eight A40 and four A100 GPUs slightly over a week. The computational bottleneck was not the numerical accuracy values but the collection of potential hallucinatory factors for analytical purposes, including logits and attention values for each head and layer.

## A.2 Additional Experiments and Results

**Per-object Performance.** We provide the per-object performance in the following Table 5 and Table 6.

**Autonomous Driving Case Study.** Figure 8 is a case study example from the nuScenes dataset [4] for autonomous driving. It illustrates the single and multi-object case, where each object is identified independently. The multi-object case exhibits more hallucination errors compared to the single-object case. This finding underscores the importance of studying multi-object hallucination, especially in real-world scenarios like autonomous driving, where multiple objects need to be detected accurately at the same time.

## B  Limitations, Licenses, and Risks

### B.1  Limitations

ROPE represents one of the pioneering efforts to publicly address the issue of multiple object hallucination. However, we acknowledge several limitations in our work: (1) The lack of transparency in the LVLMs makes it difficult to guarantee that our unseen dataset has not been previously exposed. (2) Our evaluation benchmark uses a fixed set of semantic objects, which may introduce bias and impose unnecessary constraints on the LVLMs' ability to follow instructions and reason effectively. (3) The evaluation process can be slow, as it involves performing five inferences per image for both student forcing and teacher forcing.

### B.2  Artifacts and licenses

We report a list of licenses for all datasets and models used in our experiment in Table 4. We strictly follow all the model licenses and limit the scope of these models to academic research only.

| Data Sources | URL | License |
|---|---|---|
| MSCOCO 2017 | Link | CC BY 4.0 |
| ADE20K | Link | BSD-3-Clause |

| Software Code | URL | License |
|---|---|---|
| LLaVA | Link | Llama Community Licence |
| Qwen-VL | Link | Tongyi Qianwen Licence |
| CogVLM | Link | CogVLM Licence |
| IDEFICS | Link | Llama Community Licence |
| Yi-VL | Link | Yi Community Licence |
| MiniCPM-V | Link | Apache License 2.0 |
| GLAMM | Link | Apache License 2.0 |
| GPT-4V/4O | Link | OpenAI Term of Use |

Table 4: License information for the scientific artifacts used.

### B.3  Ethical concerns and risks

This study does not require human annotators or participants for its interactive experiments. Instead, it utilizes publicly available datasets and content created by models for evaluation purposes. We are aware that these public data might introduce biases and sensitive elements, and it is essential for future research to address these concerns, possibly by creating datasets that incorporate fairness-based filtering and metrics.

| Model | Multi-Object | | | | Student Forcing | | | | Teacher Forcing | | | | Single-Object | | | |
|---|---|---|---|---|---|---|---|---|---|---|---|---|---|---|---|---|
| | Wild | Hom. | Het. | Adv. | Wild | Hom. | Het. | Adv. | Wild | Hom. | Het. | Adv. | Wild | Hom. | Het. | Adv. |
| *object 1* | | | | | | | | | | | | | | | | |
| Yi-VL-6B | 3.17 | 3.67 | 1.22 | 3.29 | 3.51 | 3.88 | 1.63 | 3.59 | 3.51 | 3.88 | 1.63 | 3.59 | 0.09 | 1.02 | 0.00 | 0.30 |
| Yi-VL-34B | 10.01 | 16.12 | 5.69 | 10.48 | 10.35 | 18.16 | 6.91 | 11.38 | 10.35 | 18.16 | 6.91 | 11.38 | 0.43 | 2.24 | 0.00 | 0.60 |
| LLaVA-7B | 34.99 | 67.14 | 13.41 | 55.39 | 34.90 | 67.14 | 13.41 | 55.39 | 34.90 | 67.14 | 13.41 | 55.39 | 34.99 | 65.71 | 15.04 | 53.89 |
| LLaVA-13B | 29.77 | 62.86 | 12.60 | 51.20 | 29.52 | 62.86 | 12.60 | 50.60 | 32.34 | 63.09 | 12.60 | 50.60 | 40.96 | 72.24 | 23.17 | 40.00 |
| LLaVA-34B | 26.61 | 77.55 | 17.48 | 62.87 | 38.58 | 88.19 | 14.23 | 60.97 | 38.58 | 88.19 | 14.23 | 60.97 | 48.33 | 77.55 | 27.64 | 66.47 |
| QwenVL | 4.01 | 2.44 | 5.92 | 4.19 | 9.56 | 20.41 | 6.91 | 16.47 | 10.49 | 20.82 | 5.28 | 17.96 | 18.39 | 32.04 | 9.76 | 19.76 |
| QwenVL-C | 22.53 | 28.16 | 13.41 | 24.25 | 22.61 | 33.88 | 13.41 | 24.25 | 23.21 | 32.86 | 8.94 | 23.65 | 34.13 | 59.80 | 16.67 | 47.01 |
| CogVLM | 1.20 | 0.00 | 0.00 | 0.00 | 0.00 | 0.00 | 0.00 | 0.00 | 0.00 | 0.00 | 0.00 | 0.00 | 0.00 | 0.00 | 0.00 | 0.00 |
| CogVLM-C | 0.00 | 0.00 | 0.00 | 0.00 | 8.94 | 1.47 | 9.76 | 7.02 | 10.16 | 16.50 | 4.17 | 10.71 | 11.83 | 13.97 | 8.94 | 7.78 |
| CogVLM-G | 17.88 | 27.76 | 7.72 | 27.54 | 23.18 | 44.90 | 9.76 | 33.83 | 23.18 | 44.90 | 9.76 | 33.83 | 30.89 | 50.82 | 10.98 | 42.22 |
| CogVLM-2 | 24.55 | 36.33 | 16.26 | 31.44 | 24.89 | 37.14 | 18.70 | 44.61 | 24.89 | 37.14 | 18.70 | 44.61 | 10.85 | 31.43 | 3.25 | 27.25 |
| IDEFICS | 1.45 | 1.63 | 1.22 | 1.50 | 11.95 | 24.29 | 6.10 | 14.67 | 11.95 | 24.29 | 6.10 | 14.67 | 4.86 | 3.88 | 5.28 | 2.10 |
| MiniCPM-V | 33.86 | 63.21 | 18.62 | 55.69 | 35.59 | 65.31 | 15.04 | 54.79 | 35.59 | 65.31 | 15.04 | 54.79 | 25.15 | 47.76 | 12.60 | 37.13 |
| LLaVA-7B* | N/A | N/A | N/A | N/A | 7.08 | 11.43 | 6.50 | 8.68 | N/A | N/A | N/A | N/A | 13.45 | 25.75 | 13.14 | 24.40 |
| GLaMM* | N/A | N/A | N/A | N/A | 53.50 | 50.20 | 41.06 | 50.30 | N/A | N/A | N/A | N/A | 68.34 | 77.55 | 54.07 | 73.35 |
| GroundHOG* | N/A | N/A | N/A | N/A | 15.27 | 20.41 | 10.98 | 19.76 | N/A | N/A | N/A | N/A | 40.10 | 28.98 | 31.30 | 32.34 |
| GPT-4V† | 49.53 | 67.35 | 38.21 | 56.29 | N/A | N/A | N/A | N/A | N/A | N/A | N/A | N/A | 47.05 | 64.49 | 36.59 | 57.19 |
| GPT-4O† | 64.42 | 80.61 | 56.10 | 73.05 | N/A | N/A | N/A | N/A | N/A | N/A | N/A | N/A | 63.56 | 81.22 | 53.66 | 73.65 |
| *object 2* | | | | | | | | | | | | | | | | |
| Yi-VL-6B | 2.65 | 4.08 | 2.44 | 3.29 | 1.88 | 8.37 | 4.47 | 5.69 | 3.68 | 7.96 | 5.69 | 9.58 | 0.17 | 0.61 | 0.00 | 0.00 |
| Yi-VL-34B | 8.55 | 17.55 | 3.25 | 15.57 | 10.61 | 19.59 | 8.54 | 12.28 | 11.21 | 22.24 | 10.98 | 14.37 | 0.51 | 2.45 | 0.41 | 0.60 |
| LLaVA-7B | 29.77 | 67.96 | 8.13 | 56.89 | 29.52 | 67.96 | 8.13 | 56.89 | 32.00 | 84.69 | 5.28 | 79.34 | 34.13 | 61.02 | 15.45 | 50.30 |
| LLaVA-13B | 29.17 | 63.06 | 4.07 | 52.40 | 28.92 | 63.27 | 9.35 | 52.10 | 37.97 | 92.37 | 6.10 | 85.63 | 41.64 | 70.82 | 21.54 | 42.58 |
| LLaVA-34B | 30.89 | 79.39 | 18.29 | 68.56 | 27.56 | 90.55 | 15.45 | 65.81 | 34.40 | 91.84 | 2.85 | 68.25 | 52.18 | 78.16 | 30.08 | 68.86 |
| QwenVL | 3.84 | 2.44 | 8.98 | 6.59 | 10.24 | 23.06 | 4.07 | 20.36 | 13.05 | 37.35 | 4.07 | 33.53 | 16.94 | 27.35 | 9.35 | 20.66 |
| QwenVL-C | 23.12 | 35.71 | 10.57 | 28.74 | 17.75 | 26.12 | 7.32 | 19.46 | 22.61 | 36.33 | 11.38 | 30.24 | 33.45 | 53.88 | 13.82 | 45.81 |
| CogVLM | 0.00 | 0.00 | 0.00 | 0.30 | 0.00 | 0.00 | 0.00 | 0.00 | 0.00 | 0.00 | 0.00 | 0.00 | 0.00 | 0.00 | 0.00 | 0.00 |
| CogVLM-C | 0.00 | 0.00 | 0.00 | 0.00 | 8.12 | 1.47 | 4.88 | 7.02 | 25.90 | 93.50 | 2.08 | 94.64 | 8.39 | 10.29 | 6.91 | 6.89 |
| CogVLM-G | 16.34 | 28.37 | 4.07 | 29.64 | 15.57 | 39.80 | 4.88 | 34.13 | 21.81 | 44.90 | 2.85 | 39.52 | 30.72 | 51.43 | 14.23 | 43.71 |
| CogVLM-2 | 25.75 | 34.08 | 19.51 | 28.14 | 24.72 | 36.53 | 22.76 | 45.21 | 26.43 | 40.61 | 26.83 | 45.21 | 11.32 | 32.45 | 5.69 | 25.45 |
| IDEFICS | 0.17 | 0.20 | 0.41 | 0.60 | 11.69 | 27.76 | 4.07 | 20.06 | 31.91 | 91.43 | 2.03 | 85.93 | 4.10 | 3.67 | 6.50 | 2.10 |
| MiniCPM-V | 35.79 | 57.73 | 17.00 | 49.10 | 31.99 | 58.37 | 13.01 | 53.29 | 31.48 | 60.41 | 13.41 | 53.29 | 26.09 | 49.80 | 14.23 | 34.13 |
| LLaVA-7B* | N/A | N/A | N/A | N/A | 8.28 | 12.86 | 4.47 | 11.08 | N/A | N/A | N/A | N/A | 10.59 | 23.00 | 5.45 | 21.43 |
| GLaMM* | N/A | N/A | N/A | N/A | 23.46 | 54.69 | 6.50 | 52.99 | N/A | N/A | N/A | N/A | 69.62 | 75.92 | 49.19 | 72.46 |
| GroundHOG* | N/A | N/A | N/A | N/A | 31.31 | 27.14 | 32.52 | 27.25 | N/A | N/A | N/A | N/A | 41.13 | 29.59 | 40.65 | 33.83 |
| GPT-4V† | 48.16 | 67.55 | 36.18 | 60.18 | N/A | N/A | N/A | N/A | N/A | N/A | N/A | N/A | 47.48 | 65.51 | 32.52 | 57.78 |
| GPT-4O† | 63.48 | 80.20 | 56.50 | 73.05 | N/A | N/A | N/A | N/A | N/A | N/A | N/A | N/A | 62.62 | 80.41 | 54.07 | 72.16 |
| *object 3* | | | | | | | | | | | | | | | | |
| Yi-VL-6B | 2.91 | 3.88 | 1.22 | 2.69 | 3.68 | 3.88 | 5.69 | 5.69 | 4.02 | 8.37 | 17.89 | 15.87 | 0.17 | 0.00 | 0.41 | 0.30 |
| Yi-VL-34B | 8.30 | 15.92 | 6.10 | 14.67 | 10.95 | 17.96 | 9.35 | 11.98 | 11.46 | 22.86 | 12.60 | 15.57 | 0.43 | 2.86 | 0.41 | 0.30 |
| LLaVA-7B | 27.97 | 68.57 | 12.60 | 57.19 | 27.99 | 68.57 | 12.20 | 57.19 | 29.95 | 99.80 | 11.38 | 99.70 | 31.74 | 63.27 | 16.26 | 51.20 |
| LLaVA-13B | 25.32 | 62.86 | 8.54 | 52.40 | 25.17 | 62.86 | 8.54 | 52.10 | 36.52 | 100.00 | 13.01 | 100.00 | 43.77 | 72.65 | 24.39 | 45.16 |
| LLaVA-34B | 29.97 | 80.00 | 21.54 | 69.16 | 27.56 | 92.91 | 13.01 | 67.42 | 41.20 | 99.32 | 8.54 | 94.79 | 54.32 | 78.37 | 34.15 | 69.46 |
| QwenVL | 2.30 | 2.03 | 7.55 | 7.19 | 6.48 | 16.53 | 4.47 | 14.37 | 21.73 | 91.02 | 4.47 | 45.51 | 18.31 | 33.47 | 12.20 | 23.95 |
| QwenVL-C | 18.09 | 33.27 | 8.54 | 24.25 | 14.85 | 28.16 | 6.50 | 19.16 | 21.67 | 68.37 | 6.91 | 60.78 | 34.47 | 57.96 | 15.85 | 49.10 |
| CogVLM | 0.00 | 0.00 | 0.00 | 0.60 | 0.00 | 0.00 | 0.00 | 0.00 | 0.00 | 0.00 | 0.00 | 0.00 | 0.00 | 0.00 | 0.00 | 0.00 |
| CogVLM-C | 0.00 | 0.60 | 0.00 | 0.00 | 7.84 | 1.47 | 4.47 | 7.02 | 26.02 | 97.50 | 0.69 | 97.62 | 10.04 | 6.62 | 11.79 | 7.49 |
| CogVLM-G | 12.15 | 25.31 | 4.88 | 25.15 | 13.34 | 37.55 | 5.69 | 31.14 | 18.39 | 56.73 | 9.35 | 50.30 | 30.12 | 52.04 | 14.23 | 43.71 |
| CogVLM-2 | 19.16 | 35.31 | 16.26 | 27.84 | 24.98 | 38.98 | 24.80 | 42.81 | 27.46 | 43.47 | 32.93 | 51.50 | 10.50 | 30.00 | 6.91 | 21.56 |
| IDEFICS | 0.09 | 0.00 | 0.00 | 0.30 | 6.23 | 23.27 | 1.22 | 13.77 | 26.02 | 99.80 | 8.94 | 100.00 | 5.20 | 3.67 | 8.54 | 2.10 |
| MiniCPM-V | 35.12 | 61.06 | 13.77 | 52.10 | 28.83 | 55.71 | 14.63 | 49.70 | 29.43 | 57.14 | 15.04 | 47.90 | 26.52 | 46.53 | 14.23 | 32.93 |
| LLaVA-7B* | N/A | N/A | N/A | N/A | 8.62 | 12.65 | 4.07 | 10.18 | N/A | N/A | N/A | N/A | 10.53 | 23.00 | 9.62 | 21.43 |
| GLaMM* | N/A | N/A | N/A | N/A | 22.53 | 56.33 | 7.32 | 54.19 | N/A | N/A | N/A | N/A | 67.15 | 77.96 | 54.07 | 73.35 |
| GroundHOG* | N/A | N/A | N/A | N/A | 24.49 | 25.31 | 30.08 | 26.35 | N/A | N/A | N/A | N/A | 40.10 | 30.82 | 40.24 | 35.33 |
| GPT-4V† | 46.96 | 62.24 | 34.15 | 52.69 | N/A | N/A | N/A | N/A | N/A | N/A | N/A | N/A | 47.73 | 65.51 | 34.55 | 56.59 |
| GPT-4O† | 61.51 | 80.41 | 57.32 | 74.55 | N/A | N/A | N/A | N/A | N/A | N/A | N/A | N/A | 63.05 | 79.59 | 51.22 | 71.56 |
| *object 4* | | | | | | | | | | | | | | | | |
| Yi-VL-6B | 2.48 | 4.29 | 0.41 | 3.29 | 3.17 | 2.24 | 7.72 | 6.29 | 4.45 | 15.10 | 13.41 | 15.27 | 0.09 | 0.20 | 0.00 | 0.30 |
| Yi-VL-34B | 6.67 | 15.31 | 4.47 | 11.98 | 10.01 | 18.37 | 7.72 | 13.17 | 11.80 | 24.90 | 15.45 | 17.66 | 0.51 | 2.04 | 0.81 | 0.60 |
| LLaVA-7B | 30.37 | 68.37 | 7.32 | 56.89 | 30.20 | 68.57 | 7.32 | 56.89 | 29.95 | 100.00 | 17.89 | 100.00 | 37.64 | 66.12 | 15.85 | 57.19 |
| LLaVA-13B | 26.26 | 63.47 | 7.32 | 52.69 | 26.19 | 63.47 | 7.32 | 52.40 | 35.41 | 100.00 | 18.29 | 100.00 | 44.45 | 72.86 | 26.42 | 43.87 |
| LLaVA-34B | 30.58 | 80.20 | 18.29 | 70.36 | 31.50 | 92.13 | 13.41 | 67.10 | 35.60 | 99.32 | 6.91 | 95.26 | 50.56 | 77.14 | 28.05 | 70.06 |
| QwenVL | 2.65 | 2.03 | 8.78 | 7.19 | 5.20 | 12.45 | 4.07 | 11.08 | 9.04 | 97.35 | 3.25 | 38.32 | 18.14 | 30.41 | 8.94 | 26.65 |
| QwenVL-C | 15.44 | 29.18 | 6.50 | 23.35 | 12.54 | 25.51 | 6.91 | 18.86 | 19.97 | 72.04 | 13.41 | 63.77 | 32.85 | 40.41 | 17.48 | 49.10 |
| CogVLM | 0.00 | 0.00 | 0.00 | 0.30 | 0.00 | 0.00 | 0.00 | 0.00 | 26.60 | 47.55 | 29.67 | 57.49 | 0.00 | 0.00 | 0.00 | 0.00 |
| CogVLM-C | 0.00 | 0.00 | 0.00 | 0.00 | 7.98 | 1.47 | 4.07 | 7.02 | 28.00 | 98.50 | 2.08 | 98.81 | 11.14 | 10.29 | 11.38 | 7.78 |
| CogVLM-G | 15.31 | 25.92 | 5.69 | 26.35 | 17.19 | 42.86 | 5.28 | 36.23 | 22.75 | 60.20 | 0.81 | 60.48 | 30.29 | 55.10 | 12.60 | 43.41 |
| CogVLM-2 | 18.05 | 34.69 | 16.26 | 29.64 | 25.15 | 38.57 | 28.86 | 41.62 | 27.63 | 49.18 | 20.73 | 57.49 | 10.97 | 31.84 | 6.10 | 26.95 |
| IDEFICS | 0.09 | 0.00 | 0.00 | 0.30 | 8.70 | 23.47 | 1.22 | 13.47 | 26.88 | 99.80 | 8.13 | 100.00 | 4.95 | 3.67 | 8.13 | 2.10 |
| MiniCPM-V | 30.26 | 60.86 | 20.24 | 52.40 | 31.39 | 58.78 | 13.82 | 53.89 | 31.39 | 61.63 | 19.92 | 55.39 | 25.15 | 48.57 | 15.85 | 38.92 |
| LLaVA-7B* | N/A | N/A | N/A | N/A | 6.66 | 11.43 | 2.44 | 9.88 | N/A | N/A | N/A | N/A | 13.06 | 23.00 | 10.90 | 21.43 |
| GLaMM* | N/A | N/A | N/A | N/A | 22.27 | 56.12 | 8.54 | 54.19 | N/A | N/A | N/A | N/A | 68.17 | 76.33 | 52.85 | 74.85 |
| GroundHOG* | N/A | N/A | N/A | N/A | 19.54 | 25.31 | 25.61 | 25.75 | N/A | N/A | N/A | N/A | 39.42 | 32.45 | 36.18 | 36.83 |
| GPT-4V† | 45.42 | 64.08 | 31.71 | 50.00 | N/A | N/A | N/A | N/A | N/A | N/A | N/A | N/A | 47.82 | 66.53 | 35.77 | 53.59 |
| GPT-4O† | 63.82 | 79.80 | 57.32 | 74.85 | N/A | N/A | N/A | N/A | N/A | N/A | N/A | N/A | 64.16 | 79.59 | 53.66 | 70.96 |
| *object 5* | | | | | | | | | | | | | | | | |
| Yi-VL-6B | 2.48 | 3.47 | 0.41 | 2.10 | 3.68 | 2.86 | 6.50 | 5.39 | 4.53 | 19.18 | 14.23 | 17.66 | 0.17 | 0.41 | 0.00 | 0.00 |
| Yi-VL-34B | 5.30 | 13.27 | 1.63 | 3.29 | 9.50 | 16.12 | 7.32 | 9.88 | 11.38 | 24.29 | 14.23 | 18.26 | 0.43 | 2.24 | 0.41 | 0.30 |
| LLaVA-7B | 30.11 | 68.57 | 10.16 | 13.17 | 30.12 | 68.57 | 10.16 | 13.17 | 32.68 | 100.00 | 18.29 | 0.00 | 36.36 | 65.92 | 18.29 | 23.35 |
| LLaVA-13B | 27.63 | 63.27 | 9.35 | 10.48 | 27.22 | 63.27 | 9.35 | 10.48 | 36.01 | 100.00 | 23.98 | 0.00 | 42.49 | 71.02 | 21.54 | 21.29 |
| LLaVA-34B | 28.44 | 80.00 | 13.01 | 8.38 | 25.20 | 91.34 | 15.85 | 18.71 | 37.20 | 99.32 | 3.66 | 5.21 | 53.81 | 78.16 | 34.15 | 34.73 |
| QwenVL | 1.19 | 0.81 | 1.94 | 1.50 | 4.35 | 9.59 | 1.22 | 1.50 | 7.68 | 47.35 | 3.25 | 0.00 | 17.11 | 32.86 | 7.32 | 11.98 |
| QwenVL-C | 15.10 | 27.35 | 4.88 | 8.38 | 13.05 | 25.31 | 4.47 | 5.09 | 21.59 | 80.41 | 15.04 | 6.29 | 36.09 | 59.80 | 13.01 | 19.16 |
| CogVLM | 0.00 | 0.00 | 0.00 | 0.00 | 0.00 | 0.00 | 0.00 | 0.00 | 0.00 | 0.00 | 0.00 | 0.00 | 0.00 | 0.00 | 0.00 | 0.00 |
| CogVLM-C | 0.00 | 0.00 | 0.00 | 0.00 | 8.12 | 1.47 | 5.69 | 7.44 | 29.00 | 100.00 | 0.00 | 0.60 | 10.18 | 12.50 | 6.50 | 9.58 |
| CogVLM-G | 16.34 | 25.51 | 5.28 | 6.59 | 16.60 | 41.22 | 4.47 | 5.69 | 27.89 | 73.47 | 10.57 | 3.89 | 30.80 | 53.06 | 15.45 | 17.66 |
| CogVLM-2 | 17.71 | 34.90 | 11.38 | 12.87 | 23.44 | 38.98 | 20.73 | 12.28 | 27.29 | 59.80 | 33.74 | 12.57 | 12.02 | 28.98 | 6.91 | 5.09 |
| IDEFICS | 0.17 | 0.00 | 0.00 | 0.00 | 6.57 | 23.47 | 0.81 | 1.80 | 27.22 | 99.80 | 13.01 | 0.00 | 4.01 | 3.47 | 4.07 | 4.79 |
| MiniCPM-V | 29.76 | 56.75 | 13.36 | 19.16 | 31.05 | 56.73 | 14.23 | 14.07 | 31.48 | 60.41 | 18.29 | 17.66 | 25.24 | 46.12 | 15.04 | 17.96 |
| LLaVA-7B* | N/A | N/A | N/A | N/A | 7.34 | 12.04 | 6.91 | 4.79 | N/A | N/A | N/A | N/A | 10.79 | 23.00 | 7.69 | 10.12 |
| GLaMM* | N/A | N/A | N/A | N/A | 23.81 | 55.31 | 7.72 | 6.29 | N/A | N/A | N/A | N/A | 69.97 | 75.55 | 51.22 | 53.29 |
| GroundHOG* | N/A | N/A | N/A | N/A | 24.91 | 25.31 | 32.11 | 18.56 | N/A | N/A | N/A | N/A | 42.92 | 30.00 | 42.28 | 30.54 |
| GPT-4V† | 38.41 | 56.94 | 31.30 | 27.54 | N/A | N/A | N/A | N/A | N/A | N/A | N/A | N/A | 46.62 | 62.65 | 37.80 | 31.44 |
| GPT-4O† | 63.74 | 80.41 | 54.07 | 53.59 | N/A | N/A | N/A | N/A | N/A | N/A | N/A | N/A | 64.67 | 78.37 | 56.10 | 54.49 |

Table 5: Complete per-object results on the unseen split.

| Model | Multi-Object | | | | Student Forcing | | | | Teacher Forcing | | | | Single-Object | | | |
|---|---|---|---|---|---|---|---|---|---|---|---|---|---|---|---|---|
| | Wild | Hom. | Het. | Adv. | Wild | Hom. | Het. | Adv. | Wild | Hom. | Het. | Adv. | Wild | Hom. | Het. | Adv. |
| *object 1* | | | | | | | | | | | | | | | | |
| Yi-VL-6B | 2.92 | 5.50 | 1.92 | 4.76 | 3.12 | 5.75 | 2.56 | 6.55 | 3.12 | 20.75 | 2.56 | 6.55 | 0.06 | 0.25 | 0.00 | 0.60 |
| Yi-VL-34B | 8.51 | 15.75 | 4.81 | 13.10 | 8.71 | 16.00 | 5.13 | 11.90 | 8.71 | 16.00 | 5.13 | 11.90 | 0.19 | 2.50 | 0.00 | 0.60 |
| LLaVA-7B | 31.51 | 65.75 | 5.00 | 51.79 | 31.45 | 65.75 | 10.90 | 51.79 | 31.45 | 65.75 | 10.90 | 51.79 | 33.01 | 62.75 | 17.95 | 47.62 |
| LLaVA-13B | 28.14 | 65.83 | 10.91 | 54.17 | 28.14 | 72.00 | 7.69 | 54.76 | 28.14 | 72.00 | 7.69 | 54.76 | 41.46 | 81.25 | 24.04 | 65.48 |
| LLaVA-34B | 34.18 | 83.50 | 8.65 | 67.26 | 48.02 | 83.75 | 29.17 | 72.02 | 48.02 | 83.75 | 29.17 | 72.02 | 54.13 | 85.00 | 17.20 | 77.98 |
| QwenVL | 2.21 | 6.00 | 1.60 | 4.17 | 7.41 | 14.25 | 1.28 | 10.71 | 5.46 | 15.25 | 2.24 | 9.52 | 8.06 | 16.50 | 4.49 | 17.86 |
| QwenVL-C | 5.65 | 14.25 | 2.24 | 10.71 | 5.59 | 9.25 | 2.24 | 8.33 | 6.04 | 8.75 | 2.24 | 7.74 | 25.02 | 43.50 | 14.10 | 30.36 |
| CogVLM | 0.00 | 0.00 | 0.00 | 0.60 | 0.00 | 0.00 | 0.00 | 0.00 | 0.00 | 0.00 | 0.00 | 0.00 | 0.00 | 0.00 | 0.00 | 0.00 |
| CogVLM-C | 0.00 | 0.00 | 0.00 | 0.00 | 11.18 | 13.50 | 4.17 | 16.67 | 3.44 | 3.00 | 1.92 | 4.76 | 12.46 | 24.75 | 5.13 | 25.60 |
| CogVLM-G | 11.70 | 18.25 | 6.09 | 19.64 | 24.44 | 41.67 | 10.30 | 29.19 | 24.61 | 49.75 | 10.90 | 29.87 | 28.85 | 55.25 | 14.74 | 40.48 |
| CogVLM-2 | 18.97 | 37.00 | 13.78 | 32.14 | 35.48 | 69.50 | 10.90 | 33.33 | 35.48 | 69.50 | 10.90 | 33.33 | 18.78 | 39.25 | 9.62 | 30.36 |
| IDEFICS | 0.00 | 3.50 | 0.64 | 1.20 | 7.41 | 20.25 | 0.96 | 14.29 | 7.41 | 20.25 | 0.96 | 14.29 | 4.86 | 0.00 | 0.00 | 0.00 |
| MiniCPM-V | 32.81 | 66.25 | 15.38 | 58.33 | 31.51 | 72.00 | 13.14 | 63.69 | 31.45 | 72.25 | 13.14 | 59.52 | 25.87 | 54.00 | 15.38 | 58.93 |
| LLaVA-7B* | N/A | | | | 9.62 | 16.25 | 6.41 | 11.31 | N/A | | | | 13.45 | 25.75 | 13.14 | 24.40 |
| GLaMM* | N/A | | | | 45.61 | 44.50 | 40.38 | 42.26 | N/A | | | | 64.33 | 81.25 | 55.77 | 73.81 |
| GroundHOG* | N/A | | | | 16.11 | 24.25 | 16.03 | 22.62 | N/A | | | | 43.86 | 43.50 | 45.19 | 51.79 |
| GPT-4V† | 56.79 | 79.75 | 41.35 | 71.43 | N/A | | | | N/A | | | | 55.30 | 76.25 | 41.35 | 71.43 |
| GPT-4O† | 69.98 | 89.50 | 66.03 | 79.76 | N/A | | | | N/A | | | | 61.35 | 73.27 | 54.47 | 69.46 |
| *object 2* | | | | | | | | | | | | | | | | |
| Yi-VL-6B | 3.83 | 5.75 | 3.21 | 6.55 | 3.31 | 6.25 | 3.53 | 7.74 | 6.17 | 10.25 | 4.81 | 6.55 | 0.19 | 0.25 | 0.00 | 1.19 |
| Yi-VL-34B | 9.10 | 15.50 | 3.21 | 13.10 | 8.90 | 16.75 | 5.77 | 10.12 | 10.59 | 21.00 | 5.77 | 23.81 | 0.26 | 2.50 | 0.32 | 1.19 |
| LLaVA-7B | 32.36 | 67.50 | 15.00 | 52.98 | 32.36 | 67.25 | 12.50 | 52.98 | 35.15 | 95.00 | 12.18 | 91.07 | 36.32 | 61.75 | 20.19 | 47.62 |
| LLaVA-13B | 32.42 | 67.67 | 13.48 | 56.55 | 32.36 | 72.75 | 14.42 | 56.55 | 37.04 | 99.25 | 11.54 | 97.02 | 43.66 | 80.50 | 24.36 | 72.02 |
| LLaVA-34B | 41.26 | 85.50 | 25.32 | 70.24 | 53.15 | 85.75 | 31.41 | 73.21 | 57.05 | 98.10 | 21.84 | 94.30 | 54.52 | 85.50 | 18.10 | 77.38 |
| QwenVL | 3.38 | 6.00 | 0.96 | 3.57 | 5.78 | 15.75 | 4.49 | 12.50 | 19.36 | 57.50 | 5.13 | 42.86 | 8.45 | 14.75 | 4.81 | 12.50 |
| QwenVL-C | 9.03 | 16.50 | 9.62 | 11.31 | 5.52 | 8.75 | 6.73 | 6.55 | 16.89 | 45.25 | 9.29 | 44.05 | 26.19 | 40.00 | 15.71 | 28.57 |
| CogVLM | 0.06 | 0.00 | 0.00 | 0.60 | 0.00 | 0.00 | 0.00 | 0.00 | 0.26 | 0.00 | 0.00 | 0.00 | 0.00 | 0.00 | 0.00 | 0.00 |
| CogVLM-C | 0.00 | 0.00 | 0.00 | 0.00 | 12.12 | 13.50 | 8.65 | 16.67 | 29.43 | 94.00 | 0.00 | 94.64 | 12.38 | 22.50 | 7.69 | 18.45 |
| CogVLM-G | 12.15 | 21.75 | 6.09 | 22.02 | 25.79 | 42.83 | 7.12 | 30.87 | 27.40 | 55.75 | 16.67 | 33.56 | 31.06 | 55.00 | 18.27 | 42.86 |
| CogVLM-2 | 24.95 | 38.00 | 21.79 | 38.10 | 38.60 | 72.00 | 12.50 | 32.14 | 36.13 | 70.25 | 14.10 | 31.55 | 21.90 | 40.00 | 14.42 | 29.76 |
| IDEFICS | 0.00 | 1.00 | 0.00 | 0.60 | 5.78 | 17.75 | 0.00 | 8.33 | 22.16 | 77.75 | 12.18 | 63.10 | 4.10 | 0.00 | 0.96 | 0.00 |
| MiniCPM-V | 36.19 | 64.75 | 19.23 | 60.71 | 33.07 | 62.75 | 16.67 | 57.14 | 33.79 | 69.00 | 17.63 | 61.31 | 30.06 | 60.75 | 17.63 | 60.12 |
| LLaVA-7B* | N/A | | | | 9.10 | 17.75 | 6.41 | 13.10 | N/A | | | | 10.59 | 23.00 | 5.45 | 21.43 |
| GLaMM* | N/A | | | | 22.22 | 55.75 | 6.73 | 56.55 | N/A | | | | 64.20 | 80.25 | 56.41 | 72.62 |
| GroundHOG* | N/A | | | | 31.32 | 33.50 | 34.29 | 30.36 | N/A | | | | 43.60 | 44.25 | 41.03 | 51.19 |
| GPT-4V† | 55.69 | 78.75 | 41.35 | 73.81 | N/A | | | | N/A | | | | 56.27 | 78.50 | 37.50 | 72.62 |
| GPT-4O† | 71.28 | 89.50 | 64.10 | 82.14 | N/A | | | | N/A | | | | 60.24 | 74.90 | 54.88 | 68.56 |
| *object 3* | | | | | | | | | | | | | | | | |
| Yi-VL-6B | 2.99 | 6.25 | 2.56 | 5.95 | 3.12 | 7.00 | 4.17 | 8.93 | 5.26 | 24.75 | 6.09 | 8.33 | 0.13 | 0.25 | 0.32 | 1.19 |
| Yi-VL-34B | 9.10 | 16.00 | 1.60 | 14.88 | 9.10 | 16.75 | 2.24 | 10.71 | 11.23 | 21.75 | 4.17 | 36.31 | 0.19 | 2.50 | 0.32 | 1.19 |
| LLaVA-7B | 29.76 | 68.00 | 5.00 | 52.98 | 29.76 | 67.50 | 12.18 | 52.98 | 28.40 | 100.00 | 11.86 | 100.00 | 33.92 | 61.50 | 14.42 | 48.81 |
| LLaVA-13B | 32.75 | 68.00 | 13.18 | 58.33 | 32.68 | 73.50 | 10.90 | 57.74 | 38.27 | 100.00 | 17.63 | 100.00 | 44.90 | 82.00 | 20.19 | 67.86 |
| LLaVA-34B | 44.70 | 86.50 | 24.68 | 70.24 | 54.00 | 86.25 | 36.86 | 73.81 | 57.89 | 98.80 | 27.63 | 98.32 | 53.41 | 87.50 | 16.32 | 77.98 |
| QwenVL | 2.86 | 7.50 | 0.96 | 7.14 | 6.30 | 18.00 | 4.49 | 16.67 | 21.25 | 92.50 | 5.13 | 88.10 | 8.58 | 16.25 | 5.77 | 17.26 |
| QwenVL-C | 10.66 | 21.25 | 7.05 | 21.43 | 6.04 | 9.00 | 5.13 | 9.52 | 16.89 | 60.75 | 9.29 | 61.31 | 26.19 | 46.00 | 10.90 | 32.74 |
| CogVLM | 0.06 | 0.00 | 0.00 | 0.60 | 0.00 | 0.00 | 0.00 | 0.00 | 0.13 | 0.75 | 0.00 | 1.19 | 0.00 | 0.00 | 0.00 | 0.00 |
| CogVLM-C | 0.00 | 0.00 | 0.00 | 0.00 | 10.84 | 13.50 | 5.77 | 16.67 | 27.03 | 94.00 | 0.00 | 94.64 | 11.46 | 21.25 | 7.05 | 16.67 |
| CogVLM-G | 12.74 | 25.75 | 7.05 | 24.40 | 26.89 | 45.00 | 14.39 | 34.23 | 27.23 | 81.25 | 19.23 | 57.72 | 29.63 | 55.50 | 16.99 | 42.26 |
| CogVLM-2 | 20.27 | 36.50 | 15.71 | 36.31 | 39.12 | 72.75 | 13.14 | 32.14 | 36.45 | 74.75 | 17.31 | 49.40 | 19.36 | 34.75 | 12.18 | 31.55 |
| IDEFICS | 0.00 | 1.00 | 0.00 | 0.90 | 6.30 | 18.75 | 0.64 | 11.90 | 15.01 | 92.50 | 8.65 | 86.90 | 5.20 | 0.00 | 0.32 | 0.00 |
| MiniCPM-V | 36.19 | 68.50 | 20.51 | 61.31 | 31.38 | 60.00 | 12.50 | 52.38 | 30.60 | 68.25 | 15.71 | 54.76 | 26.15 | 53.75 | 19.87 | 95.05 |
| LLaVA-7B* | N/A | | | | 8.77 | 16.50 | 5.45 | 11.90 | N/A | | | | 10.53 | 23.00 | 9.62 | 21.43 |
| GLaMM* | N/A | | | | 22.68 | 55.25 | 5.13 | 54.17 | N/A | | | | 62.70 | 83.00 | 47.44 | 77.38 |
| GroundHOG* | N/A | | | | 24.82 | 32.75 | 24.68 | 30.36 | N/A | | | | 45.09 | 43.50 | 35.26 | 50.00 |
| GPT-4V† | 54.45 | 79.25 | 40.06 | 70.24 | N/A | | | | N/A | | | | 55.62 | 80.25 | 39.42 | 72.02 |
| GPT-4O† | 71.35 | 90.75 | 65.71 | 83.93 | N/A | | | | N/A | | | | 60.07 | 74.29 | 53.66 | 68.26 |
| *object 4* | | | | | | | | | | | | | | | | |
| Yi-VL-6B | 3.25 | 5.50 | 1.60 | 4.76 | 4.35 | 7.75 | 5.13 | 10.71 | 6.30 | 37.75 | 4.17 | 8.93 | 0.26 | 0.25 | 0.00 | 1.19 |
| Yi-VL-34B | 8.32 | 15.75 | 4.49 | 13.69 | 9.16 | 17.00 | 5.13 | 8.93 | 11.05 | 22.75 | 5.45 | 34.52 | 0.26 | 3.00 | 0.00 | 1.19 |
| LLaVA-7B | 32.42 | 68.75 | 10.00 | 59.52 | 32.42 | 68.50 | 10.90 | 54.17 | 30.28 | 100.00 | 15.06 | 100.00 | 36.84 | 63.00 | 17.31 | 49.40 |
| LLaVA-13B | 31.38 | 68.17 | 14.39 | 94.05 | 31.32 | 73.75 | 11.54 | 58.93 | 36.06 | 100.00 | 24.68 | 100.00 | 44.83 | 80.00 | 25.96 | 64.88 |
| LLaVA-34B | 40.68 | 86.75 | 19.23 | 73.21 | 54.06 | 85.25 | 34.62 | 73.81 | 57.96 | 99.10 | 25.26 | 97.65 | 56.14 | 86.75 | 22.12 | 80.95 |
| QwenVL | 2.47 | 7.25 | 0.00 | 4.76 | 5.91 | 14.75 | 2.88 | 12.50 | 22.87 | 95.50 | 7.69 | 95.24 | 9.75 | 14.25 | 6.41 | 16.07 |
| QwenVL-C | 8.58 | 16.25 | 7.05 | 16.07 | 5.46 | 8.00 | 2.88 | 7.74 | 11.96 | 56.50 | 10.26 | 54.17 | 24.82 | 38.50 | 11.22 | 28.57 |
| CogVLM | 0.00 | 0.00 | 0.00 | 0.60 | 0.00 | 0.00 | 0.00 | 0.00 | 0.06 | 1.00 | 0.00 | 0.00 | 0.00 | 0.00 | 0.00 | 0.00 |
| CogVLM-C | 0.00 | 0.00 | 0.00 | 0.00 | 10.24 | 13.50 | 9.94 | 16.67 | 26.25 | 93.75 | 0.32 | 93.45 | 9.83 | 21.25 | 9.29 | 20.83 |
| CogVLM-G | 14.10 | 24.75 | 9.62 | 27.98 | 26.09 | 45.33 | 15.30 | 34.23 | 30.91 | 86.00 | 23.40 | 62.08 | 29.89 | 57.75 | 16.99 | 44.05 |
| CogVLM-2 | 22.94 | 37.75 | 19.87 | 36.31 | 36.58 | 66.75 | 14.74 | 33.93 | 38.40 | 75.25 | 20.51 | 61.31 | 24.24 | 37.50 | 15.71 | 26.79 |
| IDEFICS | 0.00 | 1.25 | 0.00 | 0.60 | 5.91 | 18.25 | 0.64 | 11.90 | 21.12 | 95.00 | 14.74 | 93.45 | 4.95 | 23.00 | 0.00 | 0.00 |
| MiniCPM-V | 37.36 | 64.75 | 18.59 | 60.71 | 32.42 | 59.75 | 13.14 | 56.55 | 34.05 | 67.75 | 19.55 | 55.36 | 29.14 | 56.00 | 16.35 | 62.50 |
| LLaVA-7B* | N/A | | | | 8.90 | 16.75 | 5.45 | 11.31 | N/A | | | | 13.06 | 23.00 | 10.90 | 21.43 |
| GLaMM* | N/A | | | | 21.70 | 55.75 | 6.73 | 55.95 | N/A | | | | 64.91 | 80.25 | 55.45 | 67.86 |
| GroundHOG* | N/A | | | | 20.86 | 31.50 | 21.15 | 30.95 | N/A | | | | 43.99 | 42.00 | 35.58 | 50.00 |
| GPT-4V† | 53.09 | 77.25 | 43.59 | 72.62 | N/A | | | | N/A | | | | 56.40 | 78.75 | 44.87 | 74.40 |
| GPT-4O† | 72.25 | 87.00 | 68.27 | 82.14 | N/A | | | | N/A | | | | 60.67 | 72.65 | 55.69 | 70.06 |
| *object 5* | | | | | | | | | | | | | | | | |
| Yi-VL-6B | 1.75 | 5.25 | 0.64 | 1.19 | 3.31 | 7.25 | 3.53 | 8.93 | 6.37 | 37.75 | 4.17 | 0.60 | 0.32 | 0.50 | 0.32 | 0.60 |
| Yi-VL-34B | 7.47 | 13.75 | 2.56 | 3.57 | 8.97 | 15.00 | 2.88 | 5.36 | 8.86 | 17.25 | 4.17 | 4.17 | 0.19 | 2.50 | 0.00 | 0.60 |
| LLaVA-7B | 30.41 | 67.50 | 5.00 | 7.74 | 30.41 | 67.25 | 9.62 | 8.93 | 32.16 | 100.00 | 11.86 | 0.00 | 36.52 | 62.75 | 16.99 | 20.83 |
| LLaVA-13B | 33.01 | 68.50 | 11.21 | 21.43 | 32.94 | 74.25 | 13.14 | 7.74 | 35.35 | 100.00 | 18.59 | 0.00 | 40.81 | 79.25 | 25.00 | 23.81 |
| LLaVA-34B | 38.92 | 86.50 | 16.35 | 9.52 | 54.52 | 85.00 | 37.50 | 32.14 | 61.13 | 99.30 | 22.63 | 8.05 | 57.05 | 87.75 | 21.13 | 32.14 |
| QwenVL | 2.73 | 6.25 | 1.60 | 1.79 | 5.85 | 17.25 | 5.13 | 3.57 | 24.76 | 96.75 | 7.05 | 0.00 | 8.84 | 18.50 | 6.41 | 4.17 |
| QwenVL-C | 9.68 | 16.25 | 7.37 | 4.76 | 3.70 | 8.00 | 3.53 | 1.79 | 8.77 | 67.50 | 9.29 | 0.60 | 27.75 | 48.75 | 14.10 | 15.48 |
| CogVLM | 0.06 | 0.00 | 0.00 | 0.60 | 0.00 | 0.00 | 0.00 | 0.00 | 0.06 | 3.00 | 1.00 | 0.00 | 0.00 | 0.00 | 0.00 | 0.00 |
| CogVLM-C | 0.00 | 0.00 | 0.00 | 0.00 | 9.98 | 13.50 | 5.45 | 8.93 | 27.03 | 92.50 | 0.00 | 0.60 | 10.14 | 21.75 | 6.41 | 9.52 |
| CogVLM-G | 13.78 | 23.25 | 7.05 | 10.12 | 23.64 | 43.33 | 8.94 | 19.13 | 31.12 | 91.25 | 17.31 | 8.72 | 31.38 | 56.50 | 14.74 | 13.10 |
| CogVLM-2 | 20.40 | 38.50 | 15.38 | 23.21 | 35.35 | 73.25 | 12.18 | 32.74 | 39.05 | 77.75 | 24.36 | 19.64 | 22.09 | 42.25 | 16.35 | 14.88 |
| IDEFICS | 0.00 | 0.50 | 0.00 | 0.60 | 5.85 | 18.50 | 0.96 | 1.79 | 21.18 | 95.25 | 13.78 | 0.60 | 4.01 | 0.00 | 0.32 | 0.00 |
| MiniCPM-V | 31.19 | 60.00 | 13.14 | 17.26 | 29.69 | 59.50 | 12.82 | 16.07 | 30.93 | 63.00 | 17.95 | 16.67 | 25.87 | 52.25 | 15.38 | 22.02 |
| LLaVA-7B* | N/A | | | | 9.42 | 15.75 | 3.85 | 4.76 | N/A | | | | 10.79 | 23.00 | 7.69 | 10.12 |
| GLaMM* | N/A | | | | 23.33 | 55.75 | 6.09 | 4.76 | N/A | | | | 62.90 | 84.00 | 51.92 | 46.43 |
| GroundHOG* | N/A | | | | 24.76 | 32.00 | 25.00 | 26.19 | N/A | | | | 43.86 | 42.25 | 37.82 | 42.26 |
| GPT-4V† | 48.99 | 72.75 | 37.82 | 43.45 | N/A | | | | N/A | | | | 55.88 | 77.50 | 41.99 | 39.88 |
| GPT-4O† | 71.47 | 89.50 | 66.35 | 64.29 | N/A | | | | N/A | | | | 61.52 | 74.49 | 52.85 | 53.89 |

Table 6: Complete per-object results on the seen split.

