# OpenReview forum: "Multi-Object Hallucination in Vision Language Models"
_NeurIPS.cc/2024/Conference — NeurIPS 2024 poster_

### Official Review · Reviewer_uGDi · 2024-06-28

**Soundness:** 3
**Presentation:** 2
**Contribution:** 2
**Rating:** 6
**Confidence:** 4

**Summary:**

This paper proposes a benchmark of multi-object hallucinations in Large Vision Language Models (LVLMs). Specifically, this paper explores thoroughly on how LVLMs behaves when multiple objects are prompted in user instructions. They introduce Recognition-based Object Probing Evaluation (ROPE), an automated pipeline used for collection their benchmark. This paper also presents some interesting findings, including that existing LVLMs may follow shortcuts to generate hallucinated responses.

**Strengths:**

**S1: Novelty**. Studying how object tokens affect each other in LVLM inference sounds new. Most papers addressing LVLM object hallucination are focusing on single-object. Multi-object hallucination sounds a new perspective.

**S2: Thorough analysis**. This paper conducts comprehensive analysis and discuss on 9 potential impacting factors that may lead to multi-object hallucination, which are quite a lot experiments.

**S3: Interesting academic findings**. The instruction format that induces LVLM to hallucinate is interesting. This is validated by authors a quite significant factor that leads to multi-object hallucination, which can not be simply remedied by scaling up LLMs.

**Weaknesses:**

**W1: missing evaluations**. Noticing that this paper mainly focuses on object hallucination, I wonder why authors avoid evaluating published methods that address single object hallucination in LVLMs, including decoding methods like OPERA [15] or RLHF methods [43, A]. It is quite important to check how existing single-object methods perform on the proposed multi-object benchmark. However, this part is missing from this paper, which is encouraged to be presented clearly and thoroughly to support a new benchmark.

**W2: limited significance on real-world use case.** Despite interesting findings and comprehensive analysis on proposed multi-object hallucination, this paper experiments under a very specific type of inference setup. Specifically, their experiments are conducted on a specific type of instruction format (Figure 2) and limits its output tokens to objects only (line 140-150). For proposing a new benchmark, I believe this problem is worth to do some thinking. But this part is missing from this paper. Authors are suggested to discuss more on potential impact from this paper.

[A] RLHF-V: Towards Trustworthy MLLMs via Behavior Alignment from Fine-grained Correctional Human Feedback. CVPR 2024.

**Questions:**

## **Q1**

For Figure 3 (right), ground truth object for obj5 is "orange", but both LLaVA-7B and LLaVA-34B hallucinate it as an "apple".
This paper explains this phenomena in line 165-166 as follows,

> *However, they tend to make more mistakes when all tasked object classes are different or **when a new object class is introduced after multiple repeated tasks***.

It should be noted that authors use a very specific type of instruction here,

> *Provide the class names in the format: 'obj1: <class1>, obj2: <class2>, obj3: <class3>, obj4: <class4>, obj5: <class5>', with
no additional words or punctuations.*

I have a few questions for this example here.

**(1-a)** how LLaVA-7B and LLaVA-34B performs when prompted with "*Is there an orange in this image?*" This can be a baseline to this example.

**(1-b)** it should be noted that authors adopt three types of instructions in this paper (line 136-150). What type of instruction is used for this sample?

**(1-c)** noticing that in this paper LVLMs are forced to follow a special format and decode object tokens only, what do authors think about significance of such prompting?

## **Q2**

Given special instructions (Figure 2) used in this paper, a simple exploration is to train an LVLM (e.g., LLaVA) that follows such instruction well. One reason for why LVLMs performs badly is probably that, these models do not fit these special instructions very well. Authors are suggested to include such models and results accordingly. Specifically, authors can train a LLaVA model with instruction data below. For LLaVA [29] and take ground truth objects in Figure 2 as an example, this could look like,

> *A chat between a curious user and an artificial intelligence assistant. The assistant gives helpful, detailed, and polite answers to the user's questions*.

> **USER**: *<image>*.

> **USER**: *Provide the class names in the format: obj1: <class1>, obj2: <class2>, obj3: <class3>, obj4: <class4>, obj5: <class5>, with no additional words or punctuations*.

> **ASSISTANT**: *obj1: fork, obj2: knife, obj3: whisk, obj4: lemon, obj5: jar*.

This is also explored in a recent paper [A], which I think will improve this paper and make findings from this paper more convincing.

[A] Object Recognition as Next Token Prediction. CVPR 2024.

**Limitations:**

Authors include Limitations in Appendix B.1.

---

> ### Author Rebuttal · Authors · 2024-08-07
>
> Thank you for your thorough and insightful review of our paper. We are happy that you appreciate our novelty and found our analyses thorough. We addressed your questions and concerns below. If any residual concerns remain, we would be glad to discuss further. If no concerns remain, we would appreciate it if you could raise your score.
>
> ### W1: Missing Experiments.
> We have added a comparative table in our revised manuscript that clearly presents the performance of OPERA, RLHF (MiniCPM-V), and other relevant methods on our multi-object benchmark. Due to the limit of space, we kindly redirect the reviewer to the **Rebuttal Supplement PDF**.
> - Decoding algorithm (OPERA): In the default multi-object setting, OPERA marginally improves the performance of LLaVA-1.5 in some settings, especially in settings with higher object homogeneity. OPERA decreases the performance on heterogeneous tests.
> - RL-based finetuning (MiniCPM-V): MiniCPM-V shows robust performance across different settings, consistently handling various object types effectively. This approach enhances model performance in multi-object benchmarks that even surpasses single-object settings in Wild and Hom sets. Upon inspection, we found that this model demonstrates strong visual in-context learning capability and improves correct recognition when objects of the same classes are probed together.
>
> In summary, the heterogeneous test split remains challenging given recent advances in decoding and alignment.
>
> ### W2: Limited Significance on Real-world Use Case
>
> **Real-World Use Case 1: Common Scenarios in Embodied AI**
>
> Multi-object querying is particularly relevant in embodied AI applications, such as in cooking scenarios. In a typical kitchen setting, a robot might need to identify and manipulate multiple ingredients and tools simultaneously. For example, preparing a meal might require the robot to locate and retrieve a knife, cutting board, vegetables, and spices all at once. This ability to handle multiple objects at the same time enhances the robot's efficiency and effectiveness, reducing the time taken to complete tasks and improving overall performance.
>
> We further present a case study in Autonomous Driving (see Figures 1 and 2 in **Rebuttal Supplement PDF**). This case study demonstrates how our approach can be utilized in the automotive industry to enhance the accuracy and efficiency of object recognition systems in vehicles. Figures 1 and 2 are taken from the nuScenes dataset for autonomous driving. Figure 1 illustrates the single-object case, where each object is identified independently. Figure 2 demonstrates the multi-object case, where multiple objects are detected simultaneously. The multi-object case exhibits more hallucination errors compared to the single-object case. This finding underscores the importance of studying multi-object hallucination, especially in real-world scenarios like autonomous driving, where multiple objects need to be detected accurately at the same time.
>
>
> **Real-World Use Case 2: Cost and Time Efficiency**
>
> Evaluating multiple objects simultaneously, rather than querying each object individually, can significantly save both time and resources.
>
> | Type                           | Time (per 100 images) | Cost (per 100 images) |
> |-------------------------------|-----------------------|-----------------------|
> | 5 times single-object evaluation | 512s                  | $0.575                |
> | Multi-object evaluation        | 239s                  | $0.265                |
>
>
> ### Q1: Figure 3 Example
> - For Q1-a, prompting with binary inference questions "Is there an orange in this image?" has been discussed in prior work, specifically in the POPE [24]. In our paper, we focus on the multi-object setting and our goal is to compare it with a single-object setting and ensure a fair comparison. Our single-object prompts are designed to align with our multi-object prompts for controlled studies, allowing us to directly evaluate and compare the performance under both settings.
> - For Q1-b, the instruction type used for the example in Figure 3 is the default multi-object setting, the same as Figure 2.
> - For Q1-c, we utilize student-forcing and teacher-forcing techniques to mitigate the model's dependence on the output format. These techniques allow the model to focus on predicting the next class rather than conforming to a specific output structure. Furthermore, student-forcing and teacher-forcing help us analyze whether the model learns shortcuts from the text.
>
> ### Q2: Instruction Following
>
> In fact, our student/teacher forcing probing strategies are designed "to separate out errors due to following instructions, we force the model to follow the format template and decode only the object tokens for each of the five objects. Ideally, this setting allows the model to focus solely on object recognition." By decoding the object tokens conditioned on a correct template, student/teacher forcing probing strategies allow us to evaluate the model's performance in both single-object and multi-object settings more fairly, as it factors out the template dependence. See lines 140-150 and Figure 7 in the Appendix for details.
>
> We appreciate your input and the reference to the recent paper [A]. We acknowledge its relevance to our work and will discuss this paper in our revised manuscript.

---

> > ### Comment · Reviewer_uGDi · 2024-08-11
> >
> > At first, I would like to thank for authors responding to my concerns in details.
> >
> > Good to see that authors include single-object hallucination methods, including OPERA and MiniCPM-V in their global response. My major concern of **W1** has been addressed by inclusion of these two approaches. Though, authors are suggested to include more hallucination mitigation methods in future updated manuscript.
> >
> > I also recognize potential application value of multi-object hallucination in robotics and embodied AI, which addresses my concern **W2** and improves this paper.
> >
> > Further clarifications on **Q1** and **Q2** are clear, I have no further questions.
> >
> > In all, I think this paper has its novel contribution and raise my score.

---

> > > ### Author Response · Authors · 2024-08-11
> > >
> > > We’re happy to hear that we could cover all your points. Your detailed feedback was really helpful, and we’re grateful for the time you took to review our work and share your thoughts.

---

### Official Review · Reviewer_i1pJ · 2024-07-06

**Soundness:** 3
**Presentation:** 4
**Contribution:** 3
**Rating:** 6
**Confidence:** 5

**Summary:**

Previous evaluations of large vision-language model (LVLM) hallucinations have primarily focused on single objects. This paper introduces a novel hallucination evaluation benchmark named ROPE, which simultaneously assesses multiple objects within a single scene during testing. The authors present several empirical findings about multi-object hallucinations, including the statistical correlation in object distribution and its impact on hallucinatory responses.

**Strengths:**

1. The paper is well-written and clearly articulates the differences between ROPE and previous benchmarks. Especially, observations and motivations are quite reasonable.
2. Multi-object hallucination is an under-explored aspect in the LVLM hallucination scene. Also, its related findings are valuable and worth further investigation.
3. The experiments and analyses conducted are extensive and detailed.

**Weaknesses:**

1. The authors can consider adding a simple baseline method for a prospective solution for mitigating multi-object hallucinations, as a starting point.
2. Latency analysis in evaluation could be included to provide a more comprehensive analysis. For example, comparing 5 times single-object evaluation vs. multi-object evaluation. Also, especially for student/teacher forcing as they require multiple consecutive inferences. Understanding the time efficiency of the benchmark is important for practical applications.

**Questions:**

1. Do you have any initial ideas or hypotheses for mitigating multi-object hallucinations?
2. Adding a baseline methods is not a mandatory rebuttal subject, but will be a good addition to the current version.

**Limitations:**

Yes, this paper generally addressed the limitations.

---

> ### Author Rebuttal · Authors · 2024-08-07
>
> Thank you for your thoughtful feedback. We are happy that you found our paper well-written, the problem novel and under-explored, and our experiments/analyses extensive and detailed. We addressed your questions and concerns below. If any residual concerns remain, we would be glad to discuss further. If no concerns remain, we would appreciate it if you could raise your score.
>
> ### Q1. Initial ideas or hypotheses to mitigate multi-object hallucinations
>
> We present an initial ideas on addressing multi-object hallucinations from a training perspective in Section 4.3.
> Our findings indicate that LVLMs tend to hallucinate less in homogeneous cases and more in heterogeneous cases. This suggests that LVLMs may get distracted by other objects and struggle to pay attention to the referred objects. Consequently, we propose a training-free method to mitigate this issue, see details below.
>
> ### W1, Q2. Adding a baseline for multi-object hallucinations
>
> One of the most natural way to make LVLMs "pay more attention to" what is inside the bounding box is to enlarge the amount of attention spent on the bounding box region. Since our dataset comes with ground truth bounding box regions and dimensions for each image, we retrieve the corresponding patches in the ViT that contains the bounding box region, and increase the self-attention on those tokens.
>
> After some trials and errors, we found:
> - Following ATMAN [1], we keep the selected tokens’ attention the same and scale all other tokens’ attention uniformly down by 0.7. Then after softmax, those bounding box regions will naturally obtain more attention.
> - We have to freeze the attention ratio between vision and text, and manipulate the attention within visual attention. Otherwise, LVLMs output random meaningless tokens.
>
> [1] AtMan: Understanding Transformer Predictions Through Memory Efficient Attention Manipulation. Björn Deiseroth, Mayukh Deb, Samuel Weinbach, Manuel Brack, Patrick Schramowski, Kristian Kersting. NeurIPS, 2023.
>
> |               | Default Multi-Object |                      |                      | Single-Object        |                      |                      |
> |---------------|----------------------|----------------------|----------------------|----------------------|----------------------|----------------------|
> | **Models**    | **Wild**             | **Hom.**             | **Het.**             | **Wild**             | **Hom.**             | **Het.**             |
> | **Seen**                                                                                                                                                                             |
> | LLaVA-1.5                 | 21.26%               | 52.40%               | 7.69%                | 30.59%               | 60.85%               | 12.69%               |
> | LLaVA-1.5 + ATMAN-in-Box | 23.80%         | 55.60%               | 8.50%                | 32.89%               | 63.10%               | 14.85%               |
> | **Unseen**                                                                                                                                                                                          |
> | LLaVA-1.5                 | 13.96%               | 31.88%               | 3.98%                | 26.95%               | 54.41%               | 11.06%               |
> | LLaVA-1.5 + ATMAN-in-Box| 16.00%         | 35.60%               | 4.50%                | 29.00%               | 58.20%               | 13.90%               |
>
> ### W2. Latency Analysis
>
> Thank you for your valuable feedback. We appreciate your suggestion to include a latency analysis in our evaluation. In response to your comments, we have conducted additional experiments to compare the latency of five single-object evaluations with that of a multi-object evaluation.
>
> | Type                         | Time (per 100 attempts) |
> |------------------------------|-----------------------|
> | 5 times single-object evaluation | 520.34s               |
> | Multi-object evaluation               | 256.59s               |
> | Student forcing                           | 948.32s               |
> | Teacher forcing                           | 928.71s               |

---

> > ### Comment · Reviewer_i1pJ · 2024-08-11
> >
> > I appreciate the effort authors put into addressing my feedback, particularly in relation to mitigating multi-object hallucinations and conducting the latency analysis. Also, the addition of a baseline method for multi-object hallucinations is a valuable enhancement.
> >
> > Overall, I remain inclined to accept and maintain my score.

---

> > > ### Author Response · Authors · 2024-08-11
> > >
> > > It’s great to know that we were able to meet your expectations. Thank you so much for your careful review and the insightful feedback. We genuinely appreciate the time you spent helping us improve our work.

---

### Official Review · Reviewer_i299 · 2024-07-09

**Soundness:** 2
**Presentation:** 2
**Contribution:** 2
**Rating:** 5
**Confidence:** 4

**Summary:**

This paper introduce a novel multi-object hallucination evaluation task, which assess model capabilities of classifying  multiple objects given visual prompts or spatial tokens. The benchmark dataset contains both commonly seen images and unseen images regarding the instruction dataset. Evaluation results show that open-source MLLMs and even proprietary GPT-4o struggles on this task.

**Strengths:**

1. The motivation of assessing multi-object hallucination is clear.
2. The benchmark can facility the research on understanding MLLMs from a new perspective.
3. The experiments are meticulously designed and cover a broad spectrum of factors.

**Weaknesses:**

1. The writing in the experiment section, particularly in Section 5, could benefit from significant improvement.

**Questions:**

1.The legend for Figure 5 appears to be misaligned with its content, leading to confusion.  Also the discussion about Figure 5 is confusing. The analysis about (c) (f) (h) (i) are not aligned with sub-figures, while discussions about (d) (g) are aligned with the content.  (a) and (b) are not addressed in the discussion at all.
2. It is unclear which prompting strategy (bounding box or special tokens) is utilized for each model listed in Table 2. This information should be specified for clarity.
3. Typos: LLaVA-7B (unseen) student-forcing (wild) score should be in bold.

**Limitations:**

the authors adequately addressed the limitations

---

> ### Author Rebuttal · Authors · 2024-08-07
>
> We greatly appreciate the reviewer’s time and effort reviewing this paper. We thank the reviewer for the positive feedback on the motivation, novelty, and “meticulously designed” experiments. Please see our responses to your questions below.
>
> ### W1: Improvement in Writing
>
> Specifically, we describe our improvements for presentation in the experiment sections, especially Section 5.
> - **Enhanced Definitions and Explanations in Model Behaviors**: We will clarify the definitions and explanations for factors relevant to the mechanistic behaviors of the models. Specifically, we refined the descriptions and formulas for Object Token Entropy and Visual Modality Contribution to ensure they are more comprehensible.
> - **Refined Table and Figure Captions**: We will refine the table and figure captions to provide more context and detail about the settings and the content of each figure. This ensures that readers can better understand the visual data presented and the specific conditions under which the experiments were conducted.
> - **Detailed Analysis in Section 5.2**: We will rewrite and expand the section “When Do LVLMs Experience Multi-Object Hallucinations?” to provide a more detailed and clear analysis of each factor. Each factor is now thoroughly examined, with a deeper discussion of its impact on the model’s performance.
>
> We kindly redirect the reviewer to the **General Author Rebuttals** for more details.
>
> ### Q1: Legend
>
> We apologize for the confusion caused by a mistake here. The legend for Figure 5 was misaligned with its content. The hallucinated objects are represented in Yellow and the non-hallucinated are Green. We will correct these errors in the revised manuscript and ensure that the descriptions accurately reflect the results.
>
> ### Q2: Visual Prompt
>
> We apologize for any confusion regarding the visual prompting strategies in Table 2 (Lines 194-198). For all LVLMs, we experiment with the bounding box as visual prompts. Specifically for CogVLM-G, we additionally experiment with their special grounding tokens as visual prompt input, as this is their natural visual prompt format and outperforms naive bounding box prompts. We report their performance using special grounding tokens in the Table.
>
> ### Q3: Format
>
> Thank you for catching this presentation error. We have now corrected this in the revised manuscript.

---

> > ### Comment · Reviewer_i299 · 2024-08-08
> >
> > The authors may at least need to clarify the discussion about Figure 5, 6 for the consistency between the proposed analysis and the reported experimental results.
> >
> > For example, #313 states 'LVLMs tend to hallucinate objects more frequently when they are positioned away from the center', which means the object centrality (normalized distance d between object and image center, as introduced in #273) positively correlated with the hallucination rate. While the Figure 5 (d) shows the yellow distribution (i.e. the hallucination distribution as mentioned in the author rebuttal) center is left to the green distribution.
> >
> > Overall, the author response does not address my concern about the rigor of in this work from a scientific research view.

---

> > > ### Author Response · Authors · 2024-08-08
> > > **Further Clarifications by Authors**
> > >
> > > We appreciate your instant feedback! We would like to provide a more detailed clarification regarding Figures 5 (and 6) and the Object Centrality metric.
> > >
> > > We first note that the figures and findings themselves are correct, and clarify that the consistency issue arises from typos and presentation ambiguity.
> > > - **(Clarification i) Typo in the legend for Figure 5**: The hallucinated objects are represented in yellow, and the non-hallucinated objects are in green. This color convention is used throughout the paper. In the Figure 5 legend, we mistakenly flipped this for the color blocks.
> > > - **(Clarification ii) Object Centrality**: The object centrality is defined as $1-d/D$ rather than $d/D$. We realize that our wording is confusing and will clarify this in the updated version.
> > >
> > > Regarding your specific concerns:
> > >
> > > ---
> > > > The analysis about (c) (f) (h) (i) are not aligned with sub-figures...
> > >
> > > The reason for this misalignment is the flipped legend, as we clarified in **(Clarification i)**.
> > >
> > > ---
> > > > ...discussions about (d) (g) are aligned with the content...
> > >
> > > > ...#313 states 'LVLMs tend to hallucinate objects more frequently when they are positioned away from the center', which means the object centrality (normalized distance d between object and image center, as introduced in #273) positively correlated with the hallucination rate.
> > >
> > >
> > > - **Figure (d)**: Upon **(Clarification i)** and **(Clarification ii)**, we mean to define the object centrality as $1-d/D$ rather than $d/D$. In this way, the object which is closer to the center of the image has a higher object centrality value. Considering the color for (d) was reversed, it matches our analysis that “the object centrality has only a slight influence as LVLMs tend to hallucinate objects more frequently when they are positioned away from the center.”
> > > - **Figure (g)**: Figure (g) was actually discussed correctly, indicating that non hallucinated objects appear more frequently in training data. This matches our discussion that “models are less likely to hallucinate object classes that frequently appear in the training dataset.” We hope the reviewer can take a second inspection and let us know if there are other confusions.
> > >
> > > ---
> > > > ...(a) and (b) are not addressed in the discussion at all.
> > >
> > > - Figure (a) and (b) are discussed in Section 5.2. Figure (a) is discussed in Lines 309-311, Figure (b) is discussed in Lines 311-314. We analyze that “specific data factors, such as query and object homogeneity, significantly influence model performance, with increased hallucination occurring when models process images featuring multiple object classes or a variety of objects” and “the position of object tokens seems to have minimal impact".
> > >
> > > ---
> > > > The authors may at least need to clarify the discussion about Figure 5, 6 for the consistency between the proposed analysis and the reported experimental results.
> > >
> > > While we clarify the typos and ambiguities in Figure 5, we believe that Figure 6 is consistent with our proposed analysis.
> > > - For (a), there is no significant difference between the model’s predicted class and the actual class, which aligns with our statement that “although semantic salience is a key factor in determining whether a model hallucinates, it appears to have minimal impact on the prediction of hallucinated objects”.
> > > - For (b), the figure shows that the model’s predicted class has a higher training salience than the actual class, matching our analysis that ”models are more likely to hallucinate object classes that are prevalent in the training data”.
> > > - For (c), the figure shows that the position of the predicted class is listed earlier in the input prompt than the actual class, which matches our analysis that “there is a notable preference for models to hallucinate objects that are listed early in the input prompt as candidate classes”.
> > >
> > > We apologize again for the confusion caused by the mistakenly annotated figure, but we believe this might be a minor issue that we can easily fix in the updated manuscript. Our findings and analysis remain correct and valuable to the community. We hope you can take a closer look after we provide these explanations. Again, we appreciate your feedback and believe these clarifications will enhance the clarity and coherence of our findings.
> > >
> > > **Please let is know if you have any concerns that need our clarification!**

---

> > > > ### Comment · Reviewer_i299 · 2024-08-11
> > > >
> > > > The authors' responses are clear and my concerns on writing are addressed.

---

> > > > > ### Author Response · Authors · 2024-08-11
> > > > >
> > > > > We are glad that we addressed all of your concerns. Thank you very much for your thoughtful consideration and feedback. We greatly appreciate your time and effort in reviewing our work.

---

### Official Review · Reviewer_4TNS · 2024-07-13

**Soundness:** 3
**Presentation:** 3
**Contribution:** 4
**Rating:** 7
**Confidence:** 4

**Summary:**

1. The work proposes a **new benchmark evaluation method (through modifying existing datasets)** to measure if VLMs can accurately recognize multiple objects in an image simultaneously. This evaluation pipeline, designed by adding 5 bounding boxes to each image in the dataset and creating a fixed prompt, requires the model to identify which class each object within these boxes belongs to.

2. Using this benchmark dataset, the authors analyze the performance of VLMs in recognizing multiple objects simultaneously and the **factors that may lead to their hallucination**.

3. The contribution of the article lies in its **detailed analysis and insights into understanding VLMs' multi-object recognition capabilities**. Several important takeaways are:

- performance degradation with multiple object recognition: VLMs perform worse when tasked with recognizing multiple objects at once compared to single object recognition. (not convinced by the current experiment results, need more clarification from authors -> see questions below)

- using shortcut to explain hallucination behavior: When multiple objects of the same class are required to predict at the same prompt, VLMs tend to use previously recognized classes to predict subsequent object classes, leveraging shortcuts. It suggests the model simply relies on previously identified classes for subsequent predictions.

**Strengths:**

1. The task is well-defined and the evaluation metric is clear (I like the student-forcing and teacher-forcing idea): identify the object from the 5 bounding boxes at the same time.

2. Comprehensive Analysis: It thoroughly examines various factors contributing to hallucination, offering valuable insights into potential causes. It can help the readers to understand hallucination causes in VLMs.

**Weaknesses:**

1. **Unclear Benchmark Positioning**: The paper's explanation about how this benchmark differs from other datasets is quite confusing (lines 35-48). What specific capability does this benchmark target to measure compared to others? Why is image captioning verification benchmarks insufficient for multiple object consideration? Given a limited evaluation budget, why choose this dataset over others? Is current common benchmarks focus on 1) image captioning, checking if mentioned objects appear in the image and their count, and 2) image grounding, verifying if given descriptions can be located in the image? Could you explain it in more details?

2. **Writing** (minor): Some parts of the writing could be improved for better comprehension. For example, line 53 could explain earlier why the dataset is split into four sets. Readers only understand later through experiment analysis that these sets investigate if LLMs learn shortcuts by using previously predicted classes.

3. **Clarification of Shortcuts and Spurious Correlations**:  The shortcut story is very interesting, could you elaborate this part more? What are the shortcuts and spurious correlations when claiming VLMs use them? What evidence supports this conclusion? IMO Teacher forcing results and Figure 4/5a indirectly support this, but a detailed, systematic analysis in one section would be better for understanding this.

**Questions:**

3. Lines 182/183: **How are seen/unseen datasets divided**? Is it based on the classes from the Visual Genome instruction tuning dataset?

4. Table 2's single-object setup seems confusing. Is it 1) each image with 5 bounding boxes results in **5 independent sessions** with the VLM, or 2) each session asks about one object while retaining history for the next question about another object from another bounding box, or 3) every image only have 1 bounding box? Is the wild/hom/het dataset division of the single-object column different only on the bounding boxes?
   - Table 2: If it's 1), why is the single-object recognition accuracy so low, especially for LLaVA-34B? How does having 5 bounding boxes in a single-object setup impact results? For LLaVA-34B, performance for single and multiple objects is similar, which contradicts the expectation that performance would drop when asked about multiple objects simultaneously instead of focusing on single-object. Do you have other evidence supporting this claim?

5. Lines 215 and 218: Can you explain the difference between hypothesis 1) and 2)? This section is somewhat confusing.

6. Line 254: The logic here is unclear. Why does grounding tuning have little effect? Are there experimental results supporting this conclusion?

7. Figure 5: Suggest explaining **how hallucinate and non-hallucinate are calculated in the caption**, or provide a reference link to the relevant section. For example, for Figure 5f, is the process as follows: consider every object from every 5k images -> in Toal 25k objects, and classify them into hallucinate and non-hallucinate, calculate everyone's semantic salience?

8. Figures 5f/h/i: **It seems the results of hallucinate and non-hallucinate are reversed?** The descriptions seem contradictory. For instance, Figure 5h suggests that non-hallucinate objects have higher token entropy, **implying that more uncertain tokens are less likely to hallucinate.**

9. Line 298: Why consider the last generated token instead of other positions, like the answer token?

10. Figure 6: Why compare actual class and predicted class? I find it difficult to interpret due to lack information because each image should pair predicted and actual classes. Presenting them as two distributions loses this pairing information. A caption explanation of takeaways for these 3 images might help.

11. Do you think hallucinations generation are entangled with the model's ability to correctly recognize visual text information for the bounding boxes? Have you tested the model's ability to recognize bounding boxes alone?

12. (optional) Have you considered incorporating a user study where users provide bounding boxes and then answer questions? This would measure the model's capability in real-world scenarios where bounding box boundaries might be unclear.

**Limitations:**

The author has listed their limitation in the appendix B.

---

> ### Author Rebuttal · Authors · 2024-08-07
>
> # Response to Reviewer 4TNS
>
> We greatly appreciate your dedicating your valuable time to this detailed review! We address your concerns as follows.
>
> ### Weakness
>
> Due to the limited space, we kindly redirect the reviewer to the **General Author Rebuttals**.
>
> ### Q3: Seen and Unseen
> The Seen and Unseen datasets are divided based on whether the images originate from the training or test/validation sets of the original dataset. We use MSCOCO Panoptic and ADE20K datasets to build our datasets. We didn’t use Visual Genome as it does not have an official train/val split and people use the GQA splits conventionally.
>
> ### Q4: Single-Object Setup
>
> Statement (1) is correct, the setup has “each image with 5 bounding boxes results in 5 independent sessions with the VLM.” The wild/hom/het dataset division is identical to the Multi-Object setups. The purpose is to keep all irrelevant factors untouched thus the comparisons between setups are fair and controlled.
>
> > Why is the single-object recognition accuracy so low, especially for LLaVA-34B…For LLaVA-34B, performance for single and multiple objects is similar…
>
> We note that LLaVA-34B performance for single and multiple objects is similar ONLY for the situation of Seen+Heterogeneous data split. For all the other setups, performance for single is better than multiple objects. We are also aware that the performance of LLaVA-34B on Seen+Heterogeneous+Single Object is abnormal. We reran the experiments with 16 bit precision and remove model quantization, the updated number is 19.36%, which is almost the same as the previous score. Upon closer inspection, LLaVA-34B appears to memorize images encountered during pretraining and is prone to focusing on salient objects or objects with multiple occurrences.
>
> > How does having 5 bounding boxes in a single-object setup impact results?
> There is a very marginal performance increase when keeping only one bounding box compared to five. We report the results with 5 bounding boxes to keep all irrelevant factors untouched thus the comparisons between setups are fair and controlled.
>
> ### Q5: Shortcut Hypothesis
> For hypothesis 1, the model might learn to perform class-agnostic object recognition better due to the ability to perceive context through a few-shot setting. For hypothesis 2, the model might learn to recognize one specific object in a few-shot setting. We expect improved performance on the last object B in the AAAAB setup for hypothesis 1, compared to the single object setting. For hypothesis 2 we expect the performance to be roughly the same.
>
> ### Q6: Grounded Tuning
> We refer specifically to the results for CogVLM-G presented in Table 2. The model performs poorly in the default multi-class setting, but shows relatively better performance under the single-object setting and teacher-forcing setting. We hypothesize that the grounding model fails to improve performance due to either a lack of instruction-following ability. This suggests that grounding alone does not effectively reduce multi-object hallucination in our benchmark, and conversational tuning is critical. We will add more detailed experimental results to support this conclusion in the revised manuscript.
>
> ### Q7: Figure Caption Clarity
> When calculating hallucinated and non-hallucinated objects, we consider every object from all the unseen images, and classify the model’s prediction into hallucinated and non-hallucinated. We will improve the caption in the revised manuscript.
>
> ### Q8: Figure 5 Error
> Thank you for catching this bug in the plots, we realize that the legend for Figure 5 was misaligned with its content, which led to confusion. The hallucinated objects are represented in Yellow and the non-hallucinated are Green. We will correct these errors in the revised manuscript and ensure that the descriptions accurately reflect the results.
>
> ### Q9: Last Token
> This is a confusion arising from the way we described the algorithm. In our implementation, which uses a causal transformer, the model generates tokens sequentially. At each step, while generating the next token (which is the most recent token in the context of the transformer), the model attends to all previously generated tokens. This means the model’s attention mechanism considers the entire history of tokens up to that point. We apologize for any misunderstanding caused by our writing and will clarify this in the revised manuscript.
>
> ### Q10: Actual Class and Predicted Class
> Sorry for the confusion. For a model that hallucinates object A into object B, object A is the actual class, and object B is the predicted class. Indeed, presenting them as two distributions loses this one-to-one pairing information, but Zhou et al. [67] have already studied this setup and we emphasize on the statistical comparison instead.
>
> ### Q11: OCR
> Yes, we agree that our benchmark setting may entangle hallucinations with the model’s ability to correctly recognize visual text information within the bounding boxes. To address this, we conducted controlled experiments to isolate these factors. In the single-object setting, the model performed very similarly for each of the objects 1/2/3/4/5 within the same image, which indicates that its performance is stable with each visual prompt (See Appendix for the full tables). Also, since the same issue remains for both single and multi-object settings, our findings hold as the factors are controlled. Besides, our visual prompt setups mostly follow that of the set-of-mark prompting [54].
>
> ### Q12: User Studies
> Thank you for the valuable suggestion. Incorporating a user study where users provide bounding boxes and answer questions would indeed offer a meaningful evaluation of the model’s performance in real-world scenarios with potentially unclear boundaries. Due to time and resource constraints during the rebuttal process, we were unable to implement this approach. However, we consider it an excellent direction for future work and plan to explore it in subsequent research.

---

> > ### Comment · Reviewer_4TNS · 2024-08-14
> >
> > Thank you for the response and additional experiments provided. After reading, I decide to maintain the current score and evaluation.

---

> > > ### Author Response · Authors · 2024-08-14
> > >
> > > Thank you for your detailed and thoughtful feedback throughout the review process. We’re pleased that our response and additional experiments addressed your concerns.

---

### Author Rebuttal · Authors · 2024-08-07

We thank the reviewers for their detailed and thoughtful feedback. We are glad that the reviewers appreciate our motivation, task setup, analysis, and presentation. We hereby respond to the general concerns and update our experimental results.

### General 1: Benchmark Positioning (Reviewer 4TNS)

> What specific capability does this benchmark target to measure compared to others?

ROPE specifically measures object hallucination in VLMs within a **multi-object setting**, examining how models may misperceive (e.g., by inventing nonexistent objects or becoming distracted) when tasked to focus on multiple objects concurrently, and which factors cause the hallucinations.

> Why is image captioning verification benchmarks insufficient for multiple object consideration?

In comparison to image captioning benchmarks such as CHAIR, ROPE offers two main advantages:
- Captioning benchmarks don’t address grounding and has referential ambiguity. For example, in an image with multiple apples, the ability to generate “there are apples in this image” does not imply that the model can recognize each individual apple correctly. ROPE provides a clear visual prompt that reduces such ambiguity.
- ROPE employs a fixed decoding template, enabling an automatic evaluation protocol, rather than relying on LLMs as evaluators.

> Given a limited evaluation budget, why choose this dataset over others?

ROPE is designed to be token-efficient and avoids the need for additional LLMs as evaluators. The benchmark employs a fixed decoding template, saving tokens compared to image captioning benchmarks like CHAIR.

> Is current common benchmarks focus on 1) image captioning, checking if mentioned objects appear in the image and their count, and 2) image grounding, verifying if given descriptions can be located in the image? Could you explain it in more detail?

Current common benchmarks focus on either (1) or use Yes/No questions to probe the models.
- CHAIR is one of the most well-known image captioning benchmarks for hallucination. CHAIR calculates what proportion of words generated are actually in the image according to the ground truth sentences and object segmentations.
- POPE can be seen as the image grounding benchmark for hallucination. POPE formulates the evaluation of object hallucination as a binary classification that prompts LVLMs to output “Yes” or “No”, e.g. “Is there a chair in the image?”

To the best of our knowledge, none of the existing object hallucination benchmarks satisfyingly address grounding and resolve referential ambiguity mentioned above.


### General 2: Shortcut and Spurious Correlation (Review 4TNS)

Yes, the “teacher forcing results and Figure 4” are intended for investigating shortcuts. We elaborate as follows.
- We discuss Shortcuts in Section 4.2 Paragraph 3, which are simple heuristic or rule-based solutions to a problem. In the teacher-forcing setting, we found that LLaVA models score over 90% accuracy. We design an Adversarial split, in which the first four tested objects are of the same class and we probe an object of a different class for the last one (AAAAB). The model's performance on the last object B drops to nearly zero, with almost all hallucinations labeling it as A. This is in stark contrast to 23.35% if these objects are probed individually or 19.16% when these objects are placed as the first to query in multi-object settings.
- We discuss Spurious Correlations in Section 5, which are features that appear to be statistically correlated with predictions. We present a systematic study of data-specific factors, salience/frequency, and model intrinsic behaviors. When the model hallucinates object A to object B, we study how these factors contribute to which object A to be hallucinated and which object B to hallucinate.

### General 3: Writing (Reviewer 4TNS, i299)

- **Enhanced Definitions and Explanations in Model Behaviors**: We will clarify the definitions and explanations for factors relevant to the mechanistic behaviors of the models. Specifically, we refined the descriptions and formulas for Object Token Entropy and Visual Modality Contribution to ensure they are more comprehensible.
- **Refined Table and Figure Captions**: We will refine the table and figure captions to provide more context and detail about the settings and the content of each figure. This ensures that readers can better understand the visual data presented and the specific conditions under which the experiments were conducted.
- **Detailed Analysis in Section 5.2**: We will rewrite and expand the section “When Do LVLMs Experience Multi-Object Hallucinations?” to provide a more detailed and clear analysis of each factor. Each factor is now thoroughly examined, with a deeper discussion of its impact on the model’s performance.
- **Improved Explanation of Dataset Split**: We will enhance the explanation of the dataset split and its purpose earlier in the text, providing a clearer rationale for the division into four sets. This revision ensures that readers understand the motivation behind the dataset split and how it relates to investigating whether LLMs use shortcuts by leveraging previously predicted classes.


### Updated Results and Additional Experiments

**See the PDF attached**

We present a case study in the real-world autonomous driving scenario, and also additional baselines:
- ATMAN-in-Box: A simple, training-free solution that we came up with to improve multi-object hallucination;
- Decoding strategies (OPERA)
- RL-based alignment (MiniCPM-V)
- Mechanistically grounded LVLMs (GLaMM and GroundHog)

---

### Decision · Program_Chairs · 2024-09-25

**Decision:**

Accept (poster)

**Comment:**

The paper introduces a novel evaluation protocol to systematically study multi-object hallucination in LVLMs. The reviewers appreciated the paper's clear motivation, task setup, comprehensive empirical analysis, and effective rebuttal that addressed concerns regarding benchmark positioning, experimental clarity, and evaluation methods. The authors provided additional experiments and clarifications that strengthened the paper, showing the proposed method's advantages over existing benchmarks. Despite some initial concerns about writing clarity and methodological details, the authors' revisions and explanations resolved these issues. The recommendation is to accept.